# General treatment for stereo-dynamics of state-to-state chemi-ionization reactions

Stefano Falcinelli [1✉], Franco Vecchiocattivi[1] & Fernando Pirani[2]

The investigation of chemi-ionization processes provides unique information on how the reaction dynamics depend on the energy and structure of the transition state which relate to the symmetry, relative orientation of reagent/product valence electron orbitals, and selectivity of electronic rearrangements. Here we propose a theoretical approach to formulate the optical potential for $Ne^\star(^3P_{2,0})$ noble gas atom chemi-ionizations as prototype oxidation processes. We include the selective role of atomic alignment and of the electron transfer mechanism. The state-to-state reaction probability is evaluated and a unifying description of the main experimental findings is obtained. Further, we reproduce the results of recent and advanced molecular beam experiments with a state selected Ne* beam.

The selective role of electronic rearrangements within the transition state, quantified through the use of suitable operative relations, could cast light on many other chemical processes more difficult to characterize.

[1] Department of Civil and Environmental Engineering, University of Perugia, Via G. Duranti 93, 06125 Perugia, Italy. [2] Department of Chemistry, Biology and Biotechnologies, University of Perugia, Via Elce di Sotto 8, 06123 Perugia, Italy. ✉email: stefano.falcinelli@unipg.it

The control of the dynamics of chemical reactions occurring in gaseous, condensed and between different phases, is a long-standing goal of science. Several methods have been proposed to characterize the basic role of atomic-molecular orbital alignment-orientation on the microscopic evolution of elementary chemical reactions[1]. However, manifolds of parallel channels often determine the reaction dynamics: their inter-connection and individual control is still an open question[2,3]. For the processes discussed in this study[4,5], the achievement of this basic and challenging goal is made easy, since crucial selectivity details appear to be less hidden with respect to other reactions, and their possible characterization and modeling is fundamental to cast light on the behavior of more complex cases.

Chemi-ionization reactions are very common in nature. They occur when neutral reagents promote the formation of most stable ionic products and are then considered as primary steps in flames. When these processes involve a gaseous internally excited reagent species they are also named collisional autoionizations, or Penning ionization reactions. Such reactions can be considered as prototype of gas phase bimolecular oxidation processes occurring at thermal energy, and are mainly triggered when an open-shell atom $X^*$, electronically excited in a metastable state, collides with another species M (atom or molecule) forming an excited collision complex $(X \cdots M)^*$ (see Eq. 1) that spontaneously ionizes[6–9]. $(X \cdots M)^*$ is the transition state (TS) of such elementary reactions, that under thermal conditions evolves towards an oxidation process by an electron transfer occurring between M and X colliding partners. This is schematized by the left-going arrow in Eq. (2) representing the exchange of one electron from the collisional target M to the core of the metastable excited species $X^*$:

$$X^* + M \rightarrow (X \cdots M)^* \tag{1}$$

$$(X \cdots M)^* \rightarrow \left( X \xleftarrow{e^-} M \right)^* \rightarrow (X \cdots M)^+ + e^- \rightarrow \text{ionic products} \tag{2}$$

In this way the process can lead to the final $M^+$ parent ion, associated ionic aggregates, dissociated or rearranged ionization products[8,9].

Chemi-ionizations investigated in this paper are barrierless processes that play an important role in chemistry and physics of plasmas[10–12], planetary atmospheres, interstellar environments and even in basic phenomena of biology[13]. Such reactions take place also under cold and ultra-cold temperature conditions and, from a point of view of basic sciences, their detailed investigation is crucial to address the coherent control of reactivity events at low collision energy and to utilize the quantum nature of matter[14,15]. These reactions are very important in applied sciences, as for the development of mass spectrometry techniques exploiting the soft ionization of neutral species[16] and surfaces[17,18].

From a basic point of view, chemi-ionization reactions are prototype elementary oxidation processes, involving the exchange of one single electron inside the TS (see Eqs. (1) and (2)). Respect to redox reactions occurring in the liquid phase with a multi-steps microscopic mechanism that involves the exchange of solvated electrons, in our case, analyzed oxidation processes are elementary one-step reactions.

The dynamics of chemi-ionization processes are driven by an optical potential W, first introduced by Bethe in 1940[19], defined as combination of a real, $V_t$, and an imaginary, $\Gamma$, part which controls, respectively, approach and separation of colliding partners and disappearance probability of neutral reactants into final ionic products[20–23]. Specifically, W is defined as

$$W = V_t - \frac{i}{2}\Gamma \tag{3}$$

The strength of both $V_t$ and $\Gamma$ components is expected to vary with the center-of-mass separation (or intermolecular distance) R and with the relative orientation of two reagents. As a consequence, involved interaction components and related electronic rearrangements, driving the process within the collision complex with the consequent spontaneous electron ejection, must be strongly stereo-selective[24–26]. If in Eqs. (1) and (2) M is an atom, the stereo-selectivity arises exclusively from the alignment kind of half-filled orbitals within the interatomic electric field defined by quantized projections of electronic angular momentum. Detailed information on radial and angular dependences of $V_t$ and $\Gamma$ components is up to date rather limited. Moreover, although an overview of the global phenomenology, characterized at high level of microscopic detail, suggests the existence of important selectivity in the reaction dynamics, a complete and simultaneous rationalization of all experimental findings is still lacking. Therefore, important questions still remain open, concerning the modulation of the reaction probability by the relative alignment-orientation of valence atomic-molecular orbitals of reagents and products, that controls the state-to-state dependence of the processes.

As demonstrated in this paper, the accounting for in a consistent way of all mentioned open questions corresponds to consider explicitly the selective role of structure and stability of the TS, formed via collision and referred to the system in different-permitted configurations or accessible quantum states. Recently, we suggested that $V_t$ and $\Gamma$ components of the optical potential (see Eq. (3)) must be interdependent[27,28], since they simultaneously relate to electronic rearrangements and couplings within the TS[29]. In this paper we demonstrate for the first time that the knowledge of the interdependence of $V_t$ and $\Gamma$ components is essential to cast light on unknown aspects of the reaction mechanism and to fully describe in a unifying and consistent way most of experimental observations obtained in different laboratories.

The next section focuses on the W formulation adopting suitable-operative relations for $V_t$ and $\Gamma$ components which properly include their interdependence. This allowed us to develop a theoretical treatment, applicable even under state-to state reaction conditions, characterizing the electron rearrangements inside the TS with a high-level detail. The proposed methodology is applied to the prototype $Ne^*$-Kr atom-atom reaction, for which several experimental findings are available (its extension to other cases is in progress). Important validity tests are performed by comparing experimental data with predicted values of observables.

## Results and discussion

**Experimental observables and interaction potential formulation**. The proposed treatment has been stimulated by experimental findings from collision energy dependence measurements of total and partial ionization cross sections ($\sigma$)[8,30], associative to Penning ($\sigma_{as}/\sigma_{pe}$) ratios[31–34] and Penning Ionization Electron Spectra (PIES)[25,30,35] the latter representing a sort of TS spectroscopy.

This section focuses on the formulation of the manifold of adiabatic potential energy curves, representative of $V_t$ in the various reaction channels opened by atom-atom chemiionization processes, that account for all basic features of quantum states accessible to each investigated system. Identification and modeling of the leading interaction components involved permit to define nature, strength, range and selectivity of non adiabatic

effects determining the state-to-state $\Gamma$ component. This is the crucial point to highlight and define all basic features of the state-to-state stereodynamics treatment. The following sections, reporting on the prediction of the scattering quantities with their comparison with the experimental findings, provide a quantitative important test of the methodology.

For the representation of the real component $V_t$ of the optical potential, including its dependence on the separation distance $R$ and relative orientation of interacting partners, it is proper to assume that each entrance channel[35] is determined by the weighted sum of two limiting representations, whose relative role varies with $R$ (major details are given in Supplementary Methods section of the Supplementary Information). However, it is useful to emphasize that:

a. For entrance channels, at large $R$, a simple neutral-neutral representation of the associated potential energy is sufficient since the system exhibits a substantial isotropic behavior typical of an alkaline atom interacting with a noble gas partner. An Improved Lennard Jones formulation of the interaction[36] represents completely this component[36] (see further details in Supplementary Methods section of the Supplementary Information);

b. At intermediate and short $R$, the anisotropic role of the ionic core of the metastable atom is emerging. Under these conditions the representation of the interaction in the entrance channels must take into account for the anisotropic contributions due open shell "P" nature of the ionic core[37,38].

c. Also, in the exit channels the open shell nature of the atomic ion in "P" state controls the basic features of the involved interaction anisotropy.

Therefore, all guidelines extensively developed in our laboratory[38,39] suggest to describe the interaction energy, when an open "P" shell atom or ion approaches a closed shell species, as an atom $^1S_0$, by a manifold of potential energy curves, each one associated to a specific quantum state accessible to the system. The adopted representation, defined in terms of proper quantum numbers, accounts for the relative alignment (or orientation) of reagents and products permitted within the interatomic electric field which here becomes the proper quantization axis of the interacting system. Obtained curves represent effective adiabatic interaction components since including the contributions associated to pure $\Sigma$ and $\Pi$ molecular states, defined by the electronic quantum number $\Lambda = 0$ and $\Lambda = 1$ and identified as $V_\Sigma$ and $V_\Pi$, mixed by spin orbit (SO) effects. For "P" atomic species it is sufficient to employ, for a full description of its anisotropic interaction with a $^1S_0$ atom partner[38,39], a weighted sum of $V_0$ and $V_2$ Legendre-expansion radial coefficients combined with the SO splitting. Specifically, the Legendre coefficients are defined as

$$V_0 = \frac{1}{3}(V_\Sigma + 2V_\Pi) \tag{4}$$

$$V_2 = \frac{5}{3}(V_\Sigma - V_\Pi) \tag{5}$$

while the inverse formulas are simply given by

$$V_\Sigma = V_0 + \frac{2}{5}V_2 \tag{6}$$

$$V_\Pi = V_0 - \frac{1}{5}V_2 \tag{7}$$

In this way, while $V_0$ describes the spherical average interaction component, all anisotropic contributions, are directly taken into account through the use of the anisotropic term $V_2$. The last term, accounting for the quantized spatial orientation of the valence

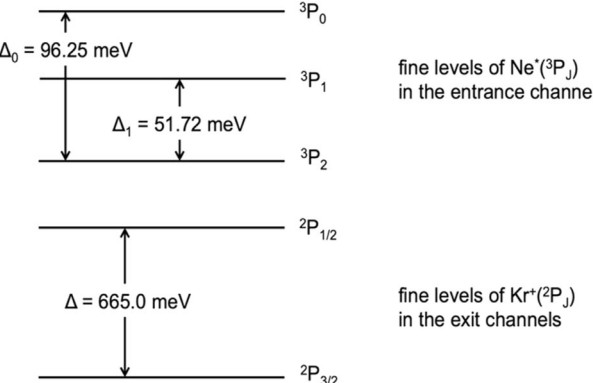

**Fig. 1 Sequence of fine levels in entrance and exit channels for Ne*-Kr system.** Note that, although Ne*($^3P_1$) is not metastable and then it does not participate to the chemi-ionization reaction, the $\Delta_1$ splitting is involved in the representation of the effective adiabatic potentials given in the text.

orbitals of the open shell atom or ion within the interacting complex, is basic to describe the manifold of adiabatic potential energy curves, associated to all quantum states accessible, with their stability anisotropy. The case of $^3P_J$ (or $^2P_J$) open shell atomic species, like Ne* (or Ne$^+$ and Kr$^+$), is characterized by a reversed sequence of SO sublevels (see Fig. 1). Accordingly, for all channels the effective adiabatic potential energy curves $V_{|J,\Omega>}$ (J is the total electronic angular momentum quantum number, defining also the spin orbit level, while $\Omega$ quantizes the absolute projection of the **J** along R) have been formulated as:

Entrance channels

$$V_{|0,0>} = V_0 + \frac{1}{10}V_2 + \frac{1}{2}\Delta_0 + \frac{1}{2}\left(\frac{9}{25}V_2^2 + \Delta_0^2 - \frac{2}{5}V_2\Delta_0\right)^{1/2} \tag{8}$$

$$V_{|2,0>} = V_0 + \frac{1}{10}V_2 + \frac{1}{2}\Delta_0 - \frac{1}{2}\left(\frac{9}{25}V_2^2 + \Delta_0^2 - \frac{2}{5}V_2\Delta_0\right)^{1/2} \tag{9}$$

$$V_{|2,1>} = V_0 + \frac{1}{10}V_2 + \frac{1}{2}\Delta_1 - \frac{1}{2}\left(\frac{9}{25}V_2^2 + \Delta_1^2\right)^{1/2} \tag{10}$$

$$V_{|2,2>} = V_0 - \frac{1}{5}V_2 \tag{11}$$

Exit channels

$$V_{|1/2,1/2>} = V_0 + \frac{1}{10}V_2 + \frac{1}{2}\Delta + \frac{1}{2}\left(\frac{9}{25}V_2^2 + \Delta^2 - \frac{2}{5}V_2\Delta\right)^{1/2} \tag{12}$$

$$V_{|3/2,1/2>} = V_0 + \frac{1}{10}V_2 + \frac{1}{2}\Delta - \frac{1}{2}\left(\frac{9}{25}V_2^2 + \Delta^2 - \frac{2}{5}V_2\Delta\right)^{1/2} \tag{13}$$

$$V_{|3/2,3/2>} = V_0 - \frac{1}{5}V_2 \tag{14}$$

where $\Delta_0$, $\Delta_1$ and $\Delta$ are the energy splitting SO between fine atomic sublevels, identified by the quantum number J, whose definition and values are given in Fig. 1.

As indicated above, in the entrance channels the $V_0$ term, coinciding with the isotropic component of $V_t(R)$, exhibits a mixed nature, accounting for the gradual passage, as $R$ decreases, from neutral-neutral to ion-neutral system surrounded by an

| ENTRANCE CHANNELS | | | | | EXIT CHANNELS | | | |
|---|---|---|---|---|---|---|---|---|
| Large R | Intermediate R | Short R | Σ character | Π character | Large-intermediate R | Short R | Σ character | Π character |
| atom-atom $|J,\Omega>$ states | ion-atom $|J,\Omega>$ states | molecular ion $^{2s+1}\Lambda_\Omega$ states | | | ion-atom $|J,\Omega>$ states | molecular ion $^{2s+1}\Lambda_\Omega$ states | | |
| $|0,0>$ | $|1/2,1/2>$ | $^2\Sigma_{1/2}$ | $C_x$ | $1-C_x$ | $|1/2,1/2>$ | $^2\Pi_{1/2}$ | $1-C_y$ | $C_y$ |
| $|2,0>$ | $|3/2,1/2>$ | $^2\Pi_{1/2}$ | 2/5 $(1-C_x)$ | 2/5 $C_x$ | $|3/2,3/2>$ | $^2\Pi_{3/2}$ | - | 1 |
| $|2,1>$ | | | 3/5 $(1-C_x)$ | 3/5 $C_x$ + 1/5 | | | | |
| $|2,2>$ | $|3/2,3/2>$ | $^2\Pi_{3/2}$ | - | 4/5 | $|3/2,1/2>$ | $^2\Sigma_{1/2}$ | $C_y$ | $1-C_y$ |

**Fig. 2 Correlation diagram between atomic and molecular states for Ne*-Kr system.** In entrance and exit channels the sequence of molecular states is opposite because the bonding and antibonding effects due to the configuration interaction. Entrance channel features: The structure of the ionic adduct, surrounded by the excited electron in a Rydberg state (see text), is of relevance to determine crucial features of the system at intermediate and short R. Entrance channels boundaries: A large R, $C_x = 0.333$ and the two fine states J = 2, 0 of Ne* exhibit Σ and Π character in the 1:2 statistical ratio. Moreover, the global weight (sum of both character degree) of $|2,0>$ state is one half of that of $|2,1>$ and $|2,2>$, because of the different degeneracy. With the R decreasing, the $|2,1>$ state assumes a Π molecular character faster than $|2,0>$ since the spin orbit mixing is characterized by $\Delta_1 < \Delta_0$ (see Fig. 1). Exit channels boundaries: At large R, $C_y = 0.667$ for the $|3/2,1/2>$ state and also in this case fine both levels exhibit Σ and Π character mixed in the 1:2 statistical ratio.

electron in a Rydberg (see left upper panel of Supplementary Fig. 1). In the exit channels, a simple ion-neutral system operates. In the two cases, the $V_0$ terms have been represented by the ILJ model (see Supplementary Methods section of the Supplementary Information), suitable to describe pure non covalent interactions.

Moreover, for both entrance and exit channels, $V_2$, that we have identified with the anisotropic configuration interaction between entrance and exit channels differing for one electron exchange (see Supplementary Fig. 2), has been represented by an exponential decreasing function, defined by a pre-exponential factor A and an exponent α. This function, defined according to the guidelines reported elsewhere[38,40] reflects the "canonical" dependence of the integral overlap between the atomic orbitals exchanging the electron. A further contribution $\frac{C_a}{R^6}$ due to the anisotropy of dispersion forces, has been also added. An important novelty of this method is that for entrance and exit channels the modulus of the exponential function is the same, while its sign is negative for exit and positive for entrance. The different sign relates to bonding and antibonding stabilization effects by charge transfer that arise from the configuration interaction between entrance and exit channels of the same symmetry[37,38,40], as depicted in Supplementary Fig. 2. The additional contribution accounts for the role of polarizability anisotropy of the open shell species on asymptotic behavior of $V_2$. All the potential parameters are given in Supplementary Table 1. The adoption of these criteria carries here to the potential energy curves consistent with the results from suitable theoretical methods on anisotropic interactions affected by the perturbation of weak chemical contributions[41,42].

Since the sign the exponential contribution to $V_2$ is positive for entrance and negative for exit channels, this interaction potential formulation leads to a different correlation between atomic states, representative of the system at long range separation distances, where $|V_2| \ll \Delta_i$, and molecular states of the same system emerging at short range, where $|V_2| \gg \Delta_i$[41].

For major details on such correlation see Fig. 2 and Supplementary Fig. 2.

Moreover, it has been also deduced that the Σ and Π character of involved potential energy curves $V_{|J,\Omega>}$ at all R values can be evaluated from the following relations[43]:

$$V_{|2,0>}, V_{|2,1>}, V_{|3/2,1/2>} = cos^2\alpha V_\Sigma + sin^2\alpha V_\Pi \quad (15)$$

$$V_{|0,0>}, V_{|1/2,1/2>} = sin^2\alpha V_\Sigma + cos^2\alpha V_\Pi \quad (16)$$

where

$$cos^2\alpha = \frac{1}{2} + \frac{\left(1 - \frac{9V_2}{5\Delta}\right)}{4\sqrt{2}\sqrt{1 + \left[\left(\frac{1-\frac{9V_2}{5\Delta}}{2\sqrt{2}}\right)\right]^2}} \quad (17)$$

Again, in all cases only the exponential contribution to the $V_2$ component with its proper sign must be taken into the previous equation. This boundary arises from the meaning of the exponential term which is the only one selectively representing the configuration interaction anisotropy by charge transfer, which is the exclusive component determining the formation of molecular states, perturbed by bonding and antibonding effects. These formulas agree with the following asymptotic conditions (see Fig. 2): at short distances, all potential energy curves must represent states having a pure Σ or Π molecular character, while at large distances, where SO coupling is dominant, a mixing of the molecular characters occurs. Note also that the adopted formulation of the interaction accounts for a different behavior of Ne*-Kr (or Ne$^+$-Kr), with respect to Ne-Kr$^+$. The variation arises from the opposite sign of $V_2$ component and from the change in the role of SO mixing, defined by the different values of $\Delta_0$, $\Delta_1$ and $\Delta$. The behavior of $V_{|2,2>}$ and $V_{|\frac{3}{2},\frac{3}{2}>}$ curves, effective in the entrance and exit channels, respectively, is not discussed in detail because they show at all distances a pure Π character. In the following, the adoption of $C_x$ and $C_y$ coefficients quantifies the Σ character degree in entrance and exit channels, respectively. Moreover, additional details on the adiabatic correlation between atomic and molecular states, both in entrance and exit channels, defined in terms of proper quantum numbers, are given in Fig. 2, where also the meaning of $C_x$ and $C_y$ coefficients, with their relevance in the definition of the suitable correlation between atomic and molecular states, is further justified. The relative role of Σ and Π molecular character in all entrance and exit channels can be also assessed from such a Figure.

From the proposed potential formulation emerges that entrance and exit channels are belonging to a unique manifold of allowed adiabatic states coupled by configuration interactions arising from the selectivity of the charge (electron) transfer (CT) component. This approach involves the use of proper quantum numbers which describe in a coherent way the passage from the atomic states (at large separation distances) to the molecular ones (emerging at short distances), both in the entrance and exit channels. As a consequence, this allows a complete representation

of the TS features and in particular a rationalization of our collected PIES data which represent a real spectroscopic probe of the reaction TS.

Here, we extend and completely address all suggestions from recent papers of our laboratory on chemi-ionization processes[27,28]. Accordingly, all electronic rearrangement contributions that determine the fate of reactions are properly taken into account to define simultaneously adiabatic and non-adiabatic effects, operative both in entrance and exit channels and, above all, in the TS. Moreover, while adiabatic effects influence essentially, as demonstrated above, strength and anisotropy of the real part of the optical potential, non adiabatic effects mostly control the selectivity of the imaginary part[27,28]. However, their contributions must be interdependent, since they simultaneously relate to: (a) external electronic cloud polarization, (b) changes in the electronic angular momentum couplings and (c) selectivity of charge transfer components coming from the overlap of valence orbitals. This treatment includes in the definition of $V_0$ term of entrance channels the role of polarization effects, while $C_x$ and $C_y$ coefficients are the proper marks of the transition from atomic to molecular states. In particular, the $R$ regions where $C_x$ and $C_y$ are fast varying, the decoupling of electronic angular momenta is effective and non adiabatic effects become more probable. It has been also demonstrated that non adiabatic effects, promoted by changes in the electronic angular coupling schemes manifest with the highest probability at a distance where $|V_2|$ is comparable with $\Delta_i$[39]. Finally, also the role of the selectivity of CT on non adiabatic effects must depend on the strength of $V_2$ component.

The proposed interaction formulation is here fully exploited to evaluate state-to-state reaction probabilities for a prototype atom-atom process with their effects on the stereodynamics.

**The Ne\*-Kr case.** The adiabatic potential energy curves $V_{|J,\Omega>}$, correlating the various atomic and ionic sublevels, both in entrance and exit channels, with the molecular states of different $\Sigma$ ($\Lambda = 0$) and $\Pi$ ($\Lambda = 1$) symmetry and accessible to the colliding system, are plotted for such prototype system in Supplementary Fig. 2.

The non adiabatic effects stimulate the transition from entrance to exit channels. Their nature suggests that their strength and radial dependence can be expressed as proper combinations of $C_x$ and $C_y$ coefficients. According to the definition given above, $C_x$ represents the $\Sigma$ molecular symmetry degree, associated to a specific atomic orbital alignment in the entrance channels and $C_y$ the corresponding $\Sigma$ molecular symmetry degree in the exit channels.

The Fig. 3 plots value and radial dependence of $C_x$ and $C_y$ coefficients obtained as detailed above for entrance and exit channels identified each one identified by J and $\Omega$ referring to Ne$^+$-Kr and Ne-Kr$^+$, being the states of the system effectively coupled by the charge transfer. Note that the different behavior arises from the opposite contribution of the configuration interaction. Moreover, taking into account that in all cases the $\Pi$ character degree is obtained as complement to 1 of the $\Sigma$ one and that the states $|3/2,3/2>$ exhibits a pure $\Pi$ character, it appears that globally the $\Sigma$:$\Pi$ character ratio is 1:2 at all $R$

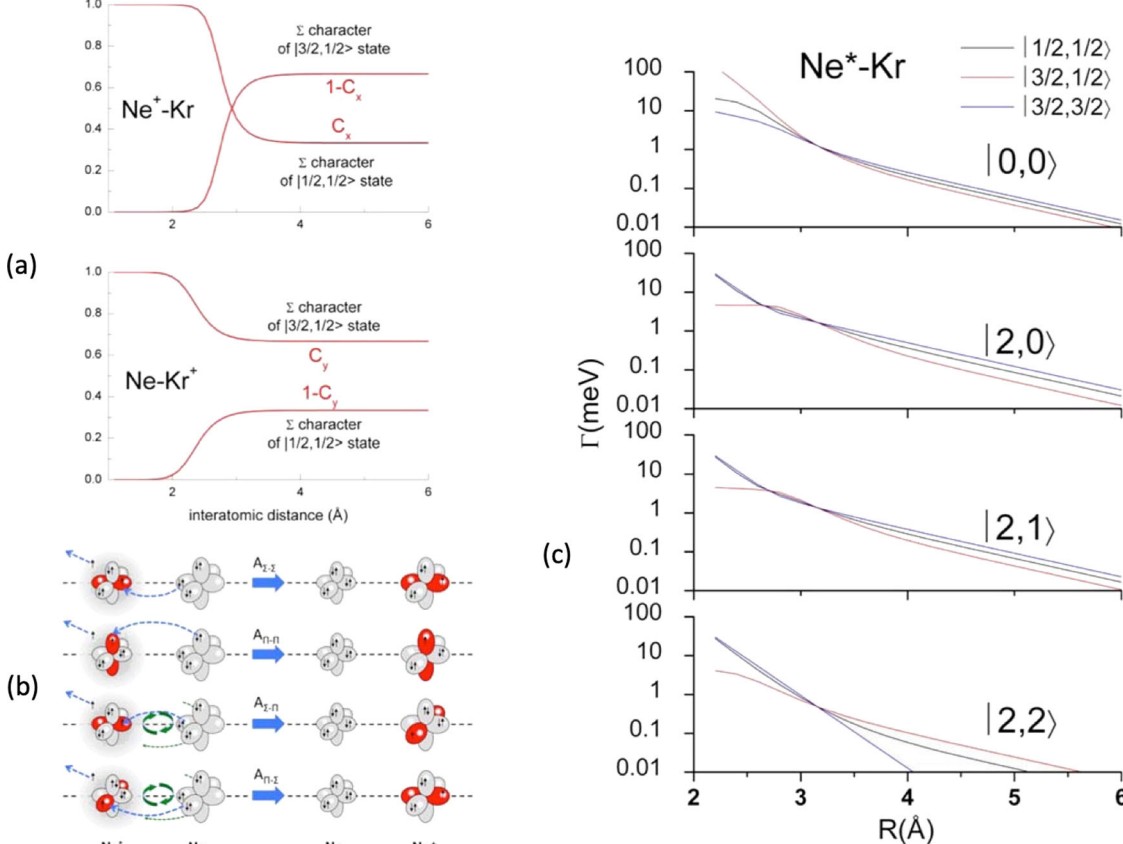

**Fig. 3 Molecular symmetry and $\Gamma$ in state-to state Ne\*-Kr chemi-ionization. a** The radial dependence of the $\Sigma$ character coefficients in entrance ($C_x$) and exit ($C_y$) channels: the $\Pi$ character is defined as complement to 1 of the $\Sigma$ one. All states accessible to the system are indicated by $|J,\Omega>$ quantum numbers. The $|3/2,3/2>$ states are not included in the Figure since they exhibit a pure $\Pi$ character at all interatomic distances. **b** A cartoon representing the main features of direct ($\Sigma$-$\Sigma$, $\Pi$-$\Pi$) and indirect ($\Sigma$-$\Pi$, $\Pi$-$\Sigma$) mechanisms of chemi-ionization processes, promoted by non-adiabatic effects operative during the collisions. **c** The state-to-state $\Gamma$ components defined in terms of $|J,\Omega>$ quantum numbers of Ne$^*$($^3P_J$) reagent and of Kr$^+$($^2P_J$) product.

according to their different degeneration. In addition, Fig. 2 provides all guidelines to obtain the $\Sigma$ to $\Pi$ character ratio for each entrance channel.

In Fig. 3 are also indicated $A_{\Lambda-\Lambda'}$, the coupling terms between entrance and exit channels, on the basis of $\Sigma$ and $\Pi$ molecular character of initial and final states of the system involved in the electron exchange. According to the criteria presented above, their value and radial dependence must be controlled by the strength $V_2$ anisotropic interaction component, by SO and Coriolis couplings.

On this ground, two different and complementary mechanisms have been also recently proposed[27,28], that are so classified:

1. Direct mechanism, promoted by nuclear vibration-electronic orbital motion couplings. It determines a homogeneous electron exchange inside the $(Ne\cdots Kr)^*$ TS of simple atom-atom chemi-ionization reactions being prototype gas-phase oxidation processes as suggested by Eqs. (1) and (2). Two different coupling terms, indicated as $A_{\Sigma-\Sigma}$ and as $A_{\Pi-\Pi}$ on the basis of the symmetry of initial and final molecular state of the system, are involved in the electron exchange (see Fig. 3). Such mechanism is expected to be more efficient at short distances where the molecular character is more prominent and/or the quantized spatial orientations of the half-filled atomic orbitals is more effective. $A_{\Sigma-\Sigma}$ is stronger than $A_{\Pi-\Pi}$ because of the different overlap integral, at the same $R$, between atomic orbitals providing molecular states of $\Sigma$ and $\Pi$ symmetry coupled by CT (see Supplementary Fig. 2).

2. Indirect mechanism, that stimulates a heterogeneous electron exchange. It is basically promoted by SO interaction (a coupling between electron magnetic moments, affected by the relativistic behavior of the spin), that provides a mixing of $\Sigma$ and $\Pi$ character, and by the nuclear rotation-electronic orbital motion interaction (Coriolis coupling). It includes also possible contributions from polarization-rearrangements of the outer electronic cloud. Moreover, from the left upper panel of Fig. 3 it appears that for interatomic distances larger than 3.5 Å the field associated to the interaction potential provides only a small perturbation on SO couplings (in particular in the entrance channel where the $C_x$ is slowly varying). However, such perturbation tends to reduce the validity of the optic selection rules. Therefore, the indirect mechanism contributions also include presumably possible contributions from radiative effects[44,45] generated by metastable atomic states perturbed within the weakly interacting complex. This mechanism involves two possible coupling terms, $A_{\Sigma-\Pi}$ and $A_{\Pi-\Sigma}$, identified again on the basis of the symmetry of the initial and final states and exhibiting a less pronounced dependence on $R$ respect to $A_{\Sigma-\Sigma}$ and $A_{\Pi-\Pi}$[27,28].

A sketch of the couplings and associated mechanisms is given in left lower panel of Fig. 3.

The important new aspect is that here we are able to evaluate the relative role of two mechanisms for each state-to-state channel with its dependence on the collision energy or on the distance range of $R$ mainly probed. In particular, the following formulations are here proposed:

$$A_{\Sigma-\Sigma}(meV) = 2.01 \times 10^6 \, e^{-4.32R} \tag{18}$$

$$A_{\Pi-\Pi}(meV) = 4.02 \times 10^5 e^{-4.32R} \tag{19}$$

$$A_{\Sigma-\Pi}(meV) = 201.0 \, e^{-1.40R} \tag{20}$$

$$A_{\Pi-\Sigma}(meV) = 40.2 \, e^{-1.40R} \tag{21}$$

where in all cases $R$ is given in Å, the first two terms represent a defined fraction of $V_2$, while the last two have been obtained adopting criteria given in refs. [27,28]. It is important to remark that overlap effects with the continuum wave function of emitted electrons[8,44] are here indirectly enclosed in the pre-exponential factor and this allows to better explicit the couplings between discrete quantum states which are more effective for the electronic rearrangements within the collision complex.

On these grounds, and considering the correlation diagram between atomic and molecular states reported in Fig. 2, we have obtained explicit relations for state-to state $\Gamma_{|J,\Omega\rightarrow J',\Omega'>}$ components, where $|J,\Omega>$ and $|J',\Omega'>$ defines initial and final quantum state, respectively. Such relations, given in the following, are formulated as weighted averages of $A_{\Lambda-\Lambda'}$ couplings, where the relative weights (their sum is normalized to 1 in each channel) are given just in terms of $C_x$ and $C_y$ coefficients:

$$\Gamma_{|0,0\rightarrow\frac{1}{2},\frac{1}{2}>} = A_{\Sigma-\Sigma}C_x\left(1-C_y\right) + A_{\Sigma-\Pi}C_xC_y + A_{\Pi-\Sigma}(1-C_x)\left(1-C_y\right) \\ + A_{\Pi-\Pi}(1-C_x)C_y \tag{22}$$

$$\Gamma_{|2,0\rightarrow\frac{1}{2},\frac{1}{2}>} = A_{\Sigma-\Sigma}(1-C_x)\left(1-C_y\right) + A_{\Sigma-\Pi}(1-C_x)C_y \\ + A_{\Pi-\Sigma}C_x\left(1-C_y\right) + A_{\Pi-\Pi}C_xC_y \tag{23}$$

$$\Gamma_{|2,1\rightarrow\frac{1}{2},\frac{1}{2}>} = A_{\Sigma-\Sigma}\frac{3}{4}(1-C_x)\left(1-C_y\right) + A_{\Sigma-\Pi}\frac{3}{4}(1-C_x)C_y \\ + A_{\Pi-\Sigma}\left(\frac{3}{4}C_x+\frac{1}{4}\right)\left(1-C_y\right) + A_{\Pi-\Pi}\left(\frac{3}{4}C_x+\frac{1}{4}\right)C_y \tag{24}$$

$$\Gamma_{|2,2\rightarrow\frac{1}{2},\frac{1}{2}>} = A_{\Pi-\Sigma}\left(1-C_y\right) + A_{\Pi-\Pi}C_y \tag{25}$$

$$\Gamma_{|0,0\rightarrow\frac{3}{2},\frac{1}{2}>} = A_{\Sigma-\Sigma}C_xC_y + A_{\Sigma-\Pi}C_x\left(1-C_y\right) + A_{\Pi-\Sigma}(1-C_x)C_y \\ + A_{\Pi-\Pi}(1-C_x)\left(1-C_y\right) \tag{26}$$

$$\Gamma_{|2,0\rightarrow\frac{3}{2},\frac{1}{2}>} = A_{\Sigma-\Sigma}(1-C_x)C_y + A_{\Sigma-\Pi}(1-C_x)\left(1-C_y\right) + A_{\Pi-\Sigma}C_xC_y \\ + A_{\Pi-\Pi}C_x\left(1-C_y\right) \tag{27}$$

$$\Gamma_{|2,1\rightarrow\frac{3}{2},\frac{1}{2}>} = A_{\Sigma-\Sigma}\frac{3}{4}(1-C_x)C_y + A_{\Sigma-\Pi}\frac{3}{4}(1-C_x)\left(1-C_y\right) \\ + A_{\Pi-\Sigma}\left(\frac{3}{4}C_x+\frac{1}{4}\right)C_y + A_{\Pi-\Pi}\left(\frac{3}{4}C_x+\frac{1}{4}\right)\left(1-C_y\right) \tag{28}$$

$$\Gamma_{|2,2\rightarrow\frac{3}{2},\frac{1}{2}>} = A_{\Pi-\Sigma}C_y + A_{\Pi-\Pi}\left(1-C_y\right) \tag{29}$$

$$\Gamma_{|0,0\rightarrow\frac{3}{2},\frac{3}{2}>} = A_{\Sigma-\Pi}C_x + A_{\Pi-\Pi}(1-C_x) \tag{30}$$

$$\Gamma_{|2,0\rightarrow\frac{3}{2},\frac{3}{2}>} = A_{\Sigma-\Pi}(1-C_x) + A_{\Pi-\Pi}C_x \tag{31}$$

$$\Gamma_{|2,1\rightarrow\frac{3}{2},\frac{3}{2}>} = A_{\Sigma-\Pi}\frac{3}{4}(1-C_x) + A_{\Pi-\Pi}\left(\frac{3}{4}C_x+\frac{1}{4}\right) \tag{32}$$

$$\Gamma_{|2,2\rightarrow\frac{3}{2},\frac{3}{2}>} = A_{\Pi-\Pi} \tag{33}$$

Obtained state-to state $\Gamma$ components are plotted in the right panel of Fig. 3. Such results suggest that at short distances, where the molecular $\Lambda$ character is more pronounced, the coupling

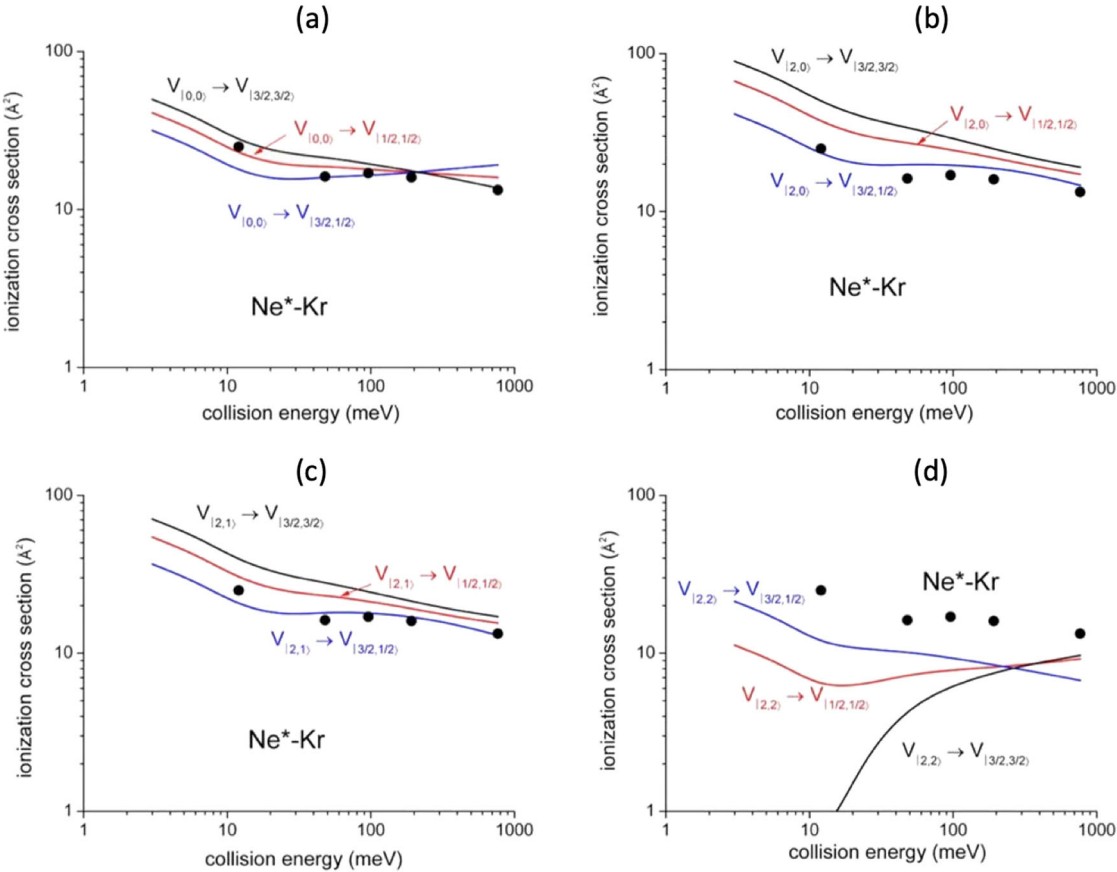

**Fig. 4 State-to-state total ionization cross sections for Ne\*-Kr.** Cross sections for: **a** |0,0> → |J',Ω'>, **b** |2,0> → |J',Ω'>, **c** |2,1> → |J',Ω'>, **d** |2,2> → |J',Ω'> state selected reaction channels (see text). The comparison with some Gregor and Siska results[45], reported as black points and referred to state averaged conditions, permits to emphasize differences in value and in energy dependence of state-to state results, while their statistical average is consistent with the experimental determination.

terms $A_{\Sigma-\Sigma}$ and $A_{\Pi-\Pi}$ become dominant, while at long range, where the molecular character is mixed by SO and other electronic arrangements operate, the indirect couplings terms, $A_{\Sigma-\Pi}$ and $A_{\Pi-\Sigma}$, become prevalent in determining the $\Gamma$ strength. Therefore, the electronic mechanisms, which drive the reaction at small and high collision energy, appear to vary substantially being preferentially triggered in the different ranges of probed $R$. Moreover, the $\Gamma$ term, evaluated in previous studies and showing only a radial dependence[45], must be related to the weighted average of state-to-state $\Gamma$ components[27,28].

**State-to-state cross sections and branching ratios.** The |J,Ω> dependence of both real and imaginary interaction components has been exploited to calculate state-to-state total (σ), Penning ($\sigma_{pe}$) and associative ($\sigma_{as}$) ionization cross sections and, as a consequence, their ratios. For the present system, Penning and associative reactions provide $Kr^+ + Ne$ and $[Kr^+ \cdots Ne]$ ionic products, respectively. Predicted state-to-state total ionization cross sections with their energy dependence are plotted in Fig. 4.

It is important to note that present calculations are able to reproduce experimental results obtained in our and other laboratories since 1981[33,45,46]. In Fig. 4, for comparison, some results of total cross section provided by Gregor and Siska[45], which are data averaged over all involved states of reagents and products, are reported. This comparison is important to emphasize both the differences between total state-to-state cross sections predicted for the various channels and to show their consistence with the absolute scale of averaged values.

Similarly, state-to-state $\sigma_{as}/\sigma_{pe}$ ratios, evaluated within the same treatment and including their energy dependence, are plotted in Fig. 5, where they are compared with state resolved results, recently obtained by Osterwalder group[33] through experiments performed with $Ne^*(^3P_2)$ reagent selected in its |J,Ω> sublevels. The exclusive purpose of such a comparison is to show again the internally consistence of predictions of our model with experimental data[33].

In the same Figure are also reported some ratios obtained in our laboratory several years ago without state selection[46]. The latter results, although referring to $\sigma_{as}/\sigma_{pe}$ ratios averaged over all permitted initial and final states, are important to provide the consistence between predicted and measured averaged ratios in an extended collision energy scale, where, after an initial increase leading to a maximum, a following pronounced fall off is observed.

In addition, in the right panel of Fig. 6 is shown a comparison between state-to-state cross section ratios, predicted by this treatment and referred to specific spin orbit sublevels of reactants and products, with peak area ratios extracted from PIESs measured as a function of the collision energy[28].

For two of the four ratios such a comparison appears to be only in semi-quantitative agreement. However, considering the difficulties involved in the separation of individual spin orbit contributions from measured energy dependence of PIESs, also for such quantities the theoretical description appears to be consistent with the experimental observables.

The proposed methodology clearly indicates how the various experimental findings depend on the basic features of real and imaginary parts of the optical potential and how their role is

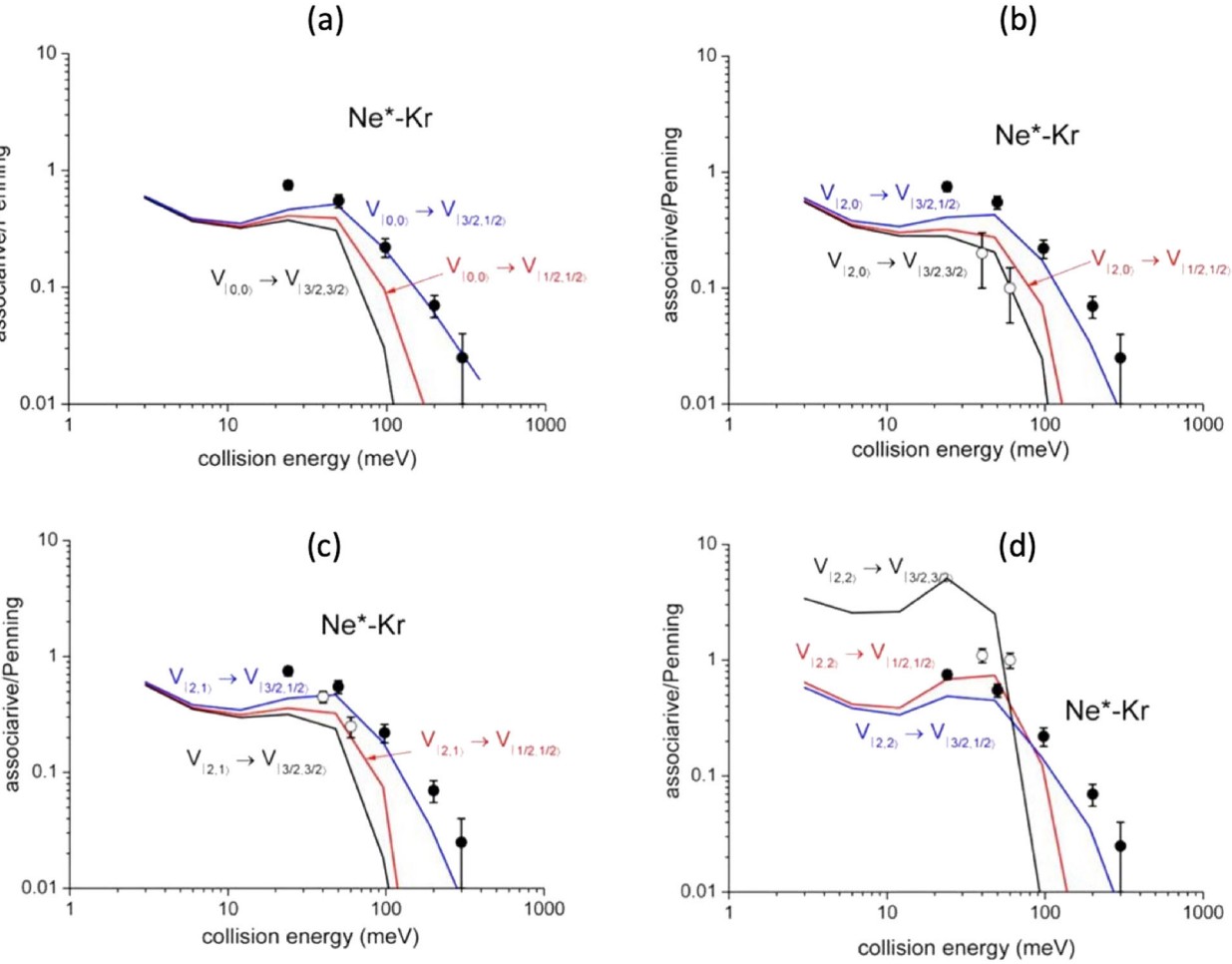

**Fig. 5 State-to-state $\sigma_{as}/\sigma_{pe}$ ratios for Ne*-Kr.** $\sigma_{as}/\sigma_{pe}$ ratios for: **a** |0,0> → |J′,Ω′>, **b** |2,0> → |J′,Ω′>, **c** |2,1> → |J′,Ω′>, **d** |2,2> → |J′,Ω′> state selected reaction channels (see text). The comparison involves some hold experimental results[46], reported as black points and referred to state averaged conditions. Recent data[33], measured with Ne*($^3P_2$) beams state selected in Ω = 2,1,0 quantum states, are also reported (open circles) for a further important comparison since they represent the unique data obtained under state selected conditions. In the figure the experimental data (black and open circles) are reported with their relative error bars.

modulated by the collision energy. In general, direct and indirect mechanisms tend to be favored, respectively, at small and high separation distance (mainly probed at high and low collision energy). From this crucial aspect it immediately emerges that strength and radial dependence of each state-to-state $\Gamma$ component depend on a different critical balance of two mechanisms. It determines respectively, value and variation with the collision energy of the total ionization cross section, which represents the reaction probability of the selected channel. Moreover, it also suggests that the $\sigma_{as}/\sigma_{pe}$ ratio mostly depends on the relative position of repulsive walls and wells in the adiabatic potential energy curves that control the dynamics of the system, including its trapping, in the entrance and exit channel of the selected state-to-state reaction (see Supplementary Fig. 2—left panel). Moreover, further basic details on observables-optical potential dependence can be emphasized and so summarized:

Low collision energy cross sections σ exhibit the highest values for entrance channels involving reagents in |2,0> and |2,1> atomic sublevels (see Fig. 4). They are affected by stronger $\Gamma$ components, reported in Fig. 3, determined by the high value of $C_x$ coefficient which exalts the role of states with major $\Sigma$ character degree at intermediate and large $R$. On the other hand, smaller values are shown by the channels with |2,2> in entrance, controlled by the exclusive $\Pi$ molecular character of

the transition states. In particular, the cross section associated to the individual elementary process |2,2>→|3/2,3/2> appears to be the lowest one, since determined exclusively by the $A_{\Pi\text{-}\Pi}$ coupling term which assumes a negligible strength at the large $R$ values probed;

High collision energy cross sections σ are slowly decreasing with the collision energy, except $\sigma_{|0,0>\rightarrow|3/2,3/2>}$, $\sigma_{|2,2>\rightarrow|1/2,1/2>}$ and $\sigma_{|2,2>\rightarrow3/2,3/2>}$ (see Fig. 4). Here the cross sections are determined by the interactions at shorter distances. Therefore, the different behavior arises from the critical balance between changes, with the $R$ decrease, of $C_x$ and $C_y$ coefficients (see Fig. 3) and of the most effective role of $A_{\Sigma\text{-}\Sigma}$ and $A_{\Pi\text{-}\Pi}$ coupling terms, promoting the direct mechanism, respect to $A_{\Sigma\text{-}\Pi}$ and $A_{\Pi\text{-}\Sigma}$ determining the indirect mechanism;

Low collision energy $\sigma_{as}/\sigma_{pe}$ ratios show the highest value for |2,2> entrance channel and this finding arises from the less repulsive wall of $V_{|2,2>}$, emerging at intermediate $R$ (see Supplementary Fig. 2), which allows a more prominent approach of reactants that favors the trapping in the potential well of the exit channels. This represents a stereodynamical feature well evident in the experimental data by Osterwalder and coworkers[33], although determined by a limited ionization probability due to the lowest total ionization cross sections (see above);

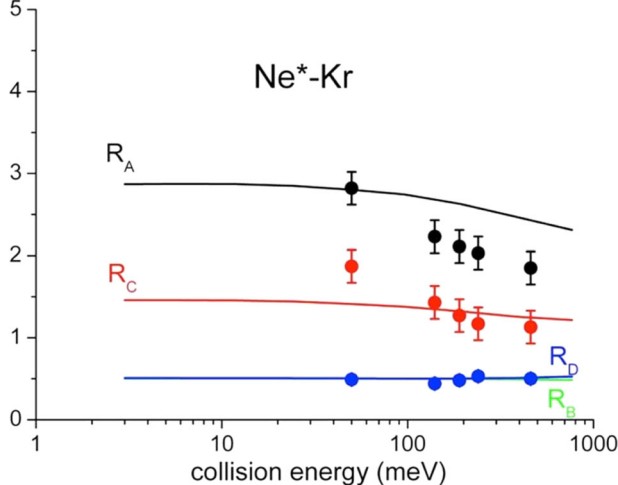

**Fig. 6 Cross section ratios of different channels |$J_i \rightarrow J_f$ > for Ne*-Kr reaction.** $R_A = \sigma_{(|2\rightarrow3/2>)}/\sigma_{(|0\rightarrow3/2>)}$; $R_B = \sigma_{(|0\rightarrow1/2>)}/\sigma_{(|0\rightarrow3/2>)}$; $R_C = \sigma_{(|2\rightarrow1/2>)}/\sigma_{(|0\rightarrow3/2>)}$; $R_D = \sigma_{(|2\rightarrow1/2>)}/\sigma_{(|2\rightarrow3/2>)}$. Points represent peak area ratios with their relative error bars extracted from the analysis of Ne*-Kr PIESs measured as a function of the collision energy, where the contributions of four reaction channels, associated to two different spin-orbit states J of Ne* neutral reactant ($J_i$ = 2, 0) and of Kr$^+$($^2P_J$) ionic product ($J_f$ = 3/2, 1/2), have been resolved (see text and Supplementary Fig. 1). The continuous lines are the results of the present treatment carried out assuming a $^3P_2$/$^3P_0$ population ratio of about 3, as found for a Ne* beam generated by electron impact[24-26].

High collision energy $\sigma_{as}/\sigma_{pe}$ ratios fall off fast, as experimentally observed[46], since the reactions provide ionic products confined in the repulsive wall of the exit channels, which lead to fully dissociated ions;

State-to-state cross section ratios, averaged over the $\Omega$ quantum numbers in order to obtain the exclusive dependence on the J spin-orbit sublevels of both neutral reactants and ionic products, are plotted as a function of the collision energy in Fig. 6 where they are indicated as $R_i$ ratios. Reported results show that the processes involving metastable atoms in J=0 highlight their relative role at higher collision energies. This effect arises from the increased value of the $C_x$ coefficient, related to an emerging more pronounced $\Sigma$ molecular character at short R. Moreover, in the probed collision energy range, the ratios referred to final states with $J_f$ = 1/2 and $J_f$ = 3/2 remain almost constant, and amount to about 0.5 since in both exit channels the $\Sigma$ and $\Pi$ molecular characters maintain always a nearly statistical weight (i.e. 1:2).

Therefore, this new methodology provides unique information on the stereo-dynamics of state-to-state chemi-ionization reactions as prototype gas-phase oxidation processes. In fact, respect to redox reactions, which occur in the liquid phase with a multi-steps microscopic mechanism that involves the exchange of solvated electrons, chemi-ionizations are elementary one-step oxidation reactions. For such processes we propose a new theoretical treatment that, in a state-to-state reaction condition, characterizes the electron rearrangements inside the TS with a high-level detail. The peculiarity of this paper consists in considering entrance and exit channels as belonging to a unique manifold of allowed quantum states coupled by configuration interactions associated to the selectivity of the electron transfer component. Moreover, also contributions from other perturbation effects are carefully taken into account. This approach involves the use of proper quantum numbers which are

describing in a consistent way the passage from the atomic states at large R to the molecular ones at short R, both in the entrance and exit channels. Adopted treatment provides a complete representation of the TS features as those affecting PIES data which represent a real spectroscopic probe of the reaction TS with its electronic rearrangements.

In this paper it is applied to a prototype atom-atom system, for which it allows to rationalize in a unifying picture most of the available experimental findings from our and other laboratories. The same type of analysis is in progress for reactions promoted by Ne*-Ar, Xe collisions. The possibility to obtain state-to-state cross sections is of great interest for the investigation of quantum effects in the coherent control of collision processes, promoting Penning and associative ionization, from ultra-cold up to thermal reactive collisions[14,15].

Moreover, the emphasized microscopic mechanisms, driven by valence electron polarization, changes in electron angular momentum couplings and by selective charge transfer effects, all operative within the collision complex, are of great relevance from a point of view of fundamental science. Specifically, they help to identify the role of electronic rearrangements determining the fate of the TS of gas-phase oxidation reactions as well as many other types of chemical processes that are more difficult to characterize. Indeed, important redox reactions occur in liquid phase with a complex multi-steps mechanism where reagents, products and exchanged electrons are solvated species. Instead, chemi-ionizations are elementary one-step gas phase reactions, involving the exchange of one single electron and this allowed us to fully characterize the electron rearrangements inside the TS with a high-level detail.

Obtained results suggest also how to extend the methodology to chemi-ionization reactions involving molecules[15], which are of great interest in several fields, including the balance of phenomena occurring in interstellar environments and planetary atmospheres. In particular, the ability of our method to describe the stereodynamics of chemi-ionization involving molecules in a state-to-state condition, should allow a correct modeling of interesting experimental data obtained by our[4] and other laboratories[12,15,47]. Finally, electron-molecule impacts promote reactions that can be considered reverse respect to autoionization processes. Such reactions are of crucial relevance in many applications of molecular plasmas (see for instance refs. [48-50] and references therein) and nuclear fusion[51].

## Methods

**Experimental technique.** All experimental data, investigated in this paper, have been measured in gas phase with the molecular beam (MB) technique applied under single collision and high resolution conditions[8,30-34]. A schematic view of MB apparatus available in our laboratory and extensively described elsewhere[27,28,35], is given in the Supplementary Methods and Supplementary Fig. 1.

**Cross section calculations.** The semiclassical method, discussed in refs. [8,9,35], has been adopted for the evaluation of all scattering properties investigated. It is applicable until the de Broglie wave length of the system assumes values shorter or comparable to 1 Å.

**Optical potential.** The real part is obtained as generalization of open shell atom interactions (see Supplementary Methods, Supplementary Fig. 2 and Supplementary Table 1). It involves both in entrance and exit channels an isotropic term, that includes non-covalent components of the interaction, and an anisotropic contribution whose "chemical" origin and role is discussed in the text (see Eqs. 4–17). Moreover, the imaginary part formulation is fully discussed in the previous section (see, in particular, Eqs. 22–33).

## Data availability

The authors declare that the data supporting the findings of this study are available within the paper and its Supplementary Information and from the corresponding author upon reasonable request.

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

## Acknowledgements

This work was supported and financed with the "Fondo Ricerca di Base, 2018, dell'Università degli Studi di Perugia" (Project Titled: Indagini teoriche e sperimentali sulla reattività di sistemi di interesse astrochimico). Support from Italian MIUR and University of Perugia (Italy) is acknowledged within the program "Dipartimenti di Eccellenza 2018-2022".

## Author contributions

S.F., F.V. and F.P. conceived and designed the study. S.F. performed experiments. S.F., F.V. and F.P. developed the proposed model with related calculations and analyzed the experimental data. All authors participated in the writing and editing the paper.

## Competing interests

The authors declare no competing interests.
