## [Peer Review File · Communications Chemistry]

Reviewers' comments:

Reviewer #1 (Remarks to the Author):

Find attached report.

Reviewer #2 (Remarks to the Author):

Report on "A new general treatment describing the stereo-dynamics of state-to-state chemi-ionization reactions as prototype of gas-phase oxidation processes" by Falcinelli et al.

This paper describes a general treatment of prototypical energy transfer reactions like Penning ionisation and related reactions. The method permits the extraction of orientation specific reactions and is applied to $\text{Ne}(3P0,2)+\text{Kr}$ as an example.

The theory is very promising and may help to greatly improve our understanding of elementary chemical reactions that play a role in many important environments. I recommend publication, but the manuscript deserves some attention before doing so. Overall, I suggest that the authors have a native-english person proof-read the manuscript in order to eliminate all the smaller linguistic mistakes and make the paper overall more easily readable. Furthermore, I have the following specific comments:

- * Line 38: "Inter-phase" should probably be interface (even if the term inter-phase is actually quite nice)
- * Line 46: I don't agree with the limitation of chemi-ionisation to reactions between excited species and something else. Chemi-ionisation reactions are also known between two ground-state species.
- * Line 55: The left-going arrow with the electron in equation 2 is not clear
- * In figure 2 (and others): The depiction of the p-orbitals in this fashion is a bit misleading and may make the reader think that the species are, in fact, oriented. While in some figures the orientation is essential, in others it is not, and the figure should probably be adapted.
- * Line 266: (X...M) can be replaced by (Ne...Kr) since in the present context the text specifically discusses that system
- * page 15/16: It is not clear what is the relation of the Gamma terms with the ratio between the AI and PI cross sections. I believe that somewhere in the paper it would be worth mentioning how one makes the step from the potentials to cross sections, especially since the readership of this article may not be experts in scattering dynamics.
- * Have the authors applied their model to the work by Kasai, Ohoyama, Watanabe et al.? This work should probably be mentioned in the paper at some point, since it experimentally investigates highly relevant systems as well.

Reviewer #3 (Remarks to the Author):

In this manuscript, the authors develop a new theoretical treatment - considering that entrance and exit channels form a unique manifold of coupled quantum states - to study state-to-state chemi-ionization reactions, and they apply it to the Ne^*-Kr system. Some parts of the manuscript are, in my opinion, quite confusing and very hard to follow, which makes me believe that the manuscript is too-specialized to warrant its publication in Communications-Chemistry.

Perhaps, the manuscript could be publishable if the authors rewrite some parts of the manuscript, focusing on the novelty of their approach (what is new in their approach).

In particular, I found very hard to understand how the computational data were calculated (in line

227 the authors say that "from a semi-empirical method" but the details are not found in the manuscript, and the authors only refer to Ref. 4 [8 articles]).

It is also not clear for me what is the meaning of 'anisotropy' in the contest of this manuscript. In line 85, the authors state that both components of W vary with R and with the relative orientation of the two atoms. However, a) it is not clear how the relative orientation is defined and, b) according to the Supporting Information, the real component only depends on R . On the same topic, it is also not clear the reduced variables on which the Legendre Polynomial depends.

I found also quite confusing the discussion about the C_x and C_y coefficients. In line 244 the authors state that the Π character degree is obtained as complement to 1 of the Σ one (given by C_x and C_y for the entrance and exit channels, respectively). However, in Fig. 3 $1-C_x$ and $1-C_y$ are also related to Σ character...

Ref. Ms. No.: COMMSCHEM-20-0037-T

Title: *A new general treatment describing the stereo-dynamics of state-to-state chemi-ionization reactions as prototype of gas phase oxidation processes.*

Authors: Stefano Falcinelli, Franco Vecchiocattivi¹ and Fernando Pirani.

I must start saying that I have read the manuscript with the greatest interest because of the subject and well established reputation of the authors.

The manuscript contains a sort of summary of the work on Penning and associative ionization carried out by the authors over the last decade. In my view, the manuscript is actually intended (perhaps unintentionally) as a review of that work which would include with some new theoretical results. Indeed, the overall treatment and most of the results have been already presented in other articles (ref. 11). Most of the panels of Figs. 1, 2 and 3 have been published before and the only truly new results (as far as I could find) are those represented in Figs. 4 and 5, where, respectively, experimental ionization cross sections and the associative/Penning cross section ratios from (mainly) other authors have been compared with the prediction of the model elaborated by the present authors.

It is not that I have anything against presenting a review in which some new results are included. I also have no objection to using previous figures to illustrate the subject. However, in my opinion, the manuscript is neither an accessible review nor a completely original self-readable article (in spite of the repetitive sentence ‘for the first time’, already present in the abstract).

The introduction, which, in effect, covers 6 pages, turns out to be too lengthy and repetitive which makes its reading somewhat tedious. In fact, part of this introduction is an advance of the the Conclusions (line 114-125 on Pg. 6 and 154-169 on Pg. 8). Similarly, the series of expansions of the imaginary part of the optical potential in terms of couplings and the C_x and C_y coefficients, can be sent to the Supporting information (SI). In contrast, some of the content of the SI would be better allocated in the main text. Indeed Figures S1 and S2 would be of great help to understand the main text.

One key issue is the semi-classical model used to simulate the experimental results portrayed in Figs. 4 and 5 which is the actual novelty of the paper. However, the authors state: ‘Although the fit of experimental data is outside the purpose of this research, which is exclusively addressed to an internally consistent prediction and rationalization of many different experimental findings ...’. Therefore, the reader is left without this information and without any reference where to find it.

As for the the reference list, I am afraid that it is not following the commonly accepted format: one reference for each paper. This has some detrimental effects such as not knowing which of the articles included under a given reference (sometimes as many as ten) have to be preferably consulted

by the interested reader. As an example, on Pg. 16, the text reads ‘...has been exploited to calculate for the first time, within a semi-classical treatment⁴...’, but, if the reader attempts to find which is the article suggested by the authors, he/she would find (to this/her dismay!) that there as many as 8 articles to peruse. In addition, there are many references that can be spared either for being too general or largely irrelevant for the content of the manuscript.

In summary, I cannot recommend the publication of this manuscript as it has been presented now. It is not self-readable or self-contained to be a review meant for illustrating the interested reader that may wish to approach the field. Many of the essential concepts to understand the manuscript are not explained at all or just in a cursory manner. The manuscript is neither a research article presenting new results (in spite of the claims by the authors); apart from Pgs. 16-20, the rest of the manuscript is written as a lengthy Introduction or anticipated Conclusions. I must say that I learnt more from ref. 11(a) than from the content of this manuscript. As for the stereodynamical content, it is somewhat hidden in the the text, and the authors do not make any attempt why that keyword is in the title.

I should stress that I am not denying the scientific content of the manuscript, even less the sound scientific reputation of their authors. I am only stating, that in my opinion, for the manuscript to be recommended for publication must be rewritten thoroughly, possibly made more concise and, for sure, more self-contained.

Point-by-point responses to the reviewers' comments - manuscript COMMSCHEM-20-0037-T.

We thank all reviewers #1, #2 and #3 for their comments and suggestions. They led to an improvement of the readability and quality of the manuscript COMMSCHEM-20-0037-T.

We have carefully and deeply revised our manuscript, fully addressing all criticisms and suggestions of the reviewers. Following the suggestions of reviewers, we rewrote our manuscript as it follows: i) the title and abstract have been synthesized (max 15 and 150 words, respectively) in order to meet the requirements of the Journal; ii) a new paragraph titled “**Experimental observables and interaction potential formulation**” has been added, moving a part of the old SI into the revised text with related necessary references; iii) the Figures 1 and 2 have been moved in SI and are replaced by the old two Figs. S1 and S2; iv) to allow this previous modification compatible with the text of the manuscript, we extracted the right lower panel of the old Figure 1 as a new Figure 6 that is inserted with the relative caption at page 22 P, of the revised version of the manuscript where the reported data are discussed; v) Nr. 18 old refs. have been deleted since not strictly necessary and all needed references have been renumbered and cited in the proper place in the text; vi) according to the article's format of the Journal, we deleted the “Conclusion” section and we added the following sentences at the end of the “Introduction” section of the revised manuscript (see top of page 6): “**In conclusion, the emphasized microscopic mechanisms, driven by valence electron polarization, changes in electron angular momentum couplings and by selective charge transfer effects, all operative within the collision complex, are of great relevance from a point of view of fundamental science. Specifically, they help to identify the role of electronic rearrangements determining the fate of the TS of gas-phase oxidation reactions as well as many other types of chemical processes that are more difficult to characterize. Indeed, important redox reactions occur in liquid phase with a complex multi-steps mechanisms where reagents, products and exchanged electrons are solvated species. Instead, chemi-ionizations are elementary one-step gas phase reactions, involving the exchange of one single electron and this allowed us to fully characterize the electron rearrangements inside the TS with a high-level detail.**”; vii) the SI has been rewritten with relative new Figures S1 and S2 (see point iii) above) and refs.

All made changes in the manuscript are reported in red color in its revised version (see new uploaded revised manuscript) and are listed below.

Responses to the Reviewer #1 comments:

Reviewer #1 – addressed point 1: *“I must start saying that I have read the manuscript with the greatest interest because of the subject and well established reputation of the authors. The manuscript contains a sort of summary of the work on Penning and associative ionization carried out by the authors over the last decade. In my view, the manuscript is actually intended (perhaps unintentionally) as a review of that work which would include with some new theoretical results. Indeed, the overall treatment and most of the results have been already presented in other articles (ref. 11). Most of the panels of Figs. 1, 2 and 3 have been published before and the only truly new results (as far as I could find) are those represented in Figs. 4 and 5, where, respectively, experimental ionization cross sections and the associative/Penning cross*

section ratios from (mainly) other authors have been compared with the prediction of the model elaborated by the present authors. It is not that I have anything against presenting a review in which some new results are included. I also have no objection to using previous figures to illustrate the subject. However, in my opinion, the manuscript is neither an accessible review nor a completely original self-readable article (in spite of the repetitive sentence 'for the first time', already present in the abstract). The introduction, which, in effect, covers 6 pages, turns out to be too lengthy and repetitive which makes its reading somewhat tedious. In fact, part of this introduction is an advance of the Conclusions (line 114-125 on Pg. 6 and 154-169 on Pg. 8). Similarly, the series of expansions of the imaginary part of the optical potential in terms of couplings and the C_x and C_y coefficients, can be sent to the Supporting information (SI). In contrast, some of the content of the SI would be better allocated in the main text. Indeed Figures S1 and S2 would be of great help to understand the main text."

Author reply and made modifications: We thank the reviewer #1 for his comment and we agree with him. In order to separate the new results discussed in the manuscript respect to those already published which are useful to the reader for a complete understand of the text, we have moved Figs. 1 and 2 with the relative text in the Supporting Information (SI). For such a reason the old Figures 1 and 2 are the new Figures S1 and S2 and the following sentences from page 5 (line 6 from the bottom) to page 8 (line 7 from the top) of the old manuscript version, have been moved to SI: "The experiments performed with the molecular beam (MB) technique provided total and partial ionization cross sections (σ),^[5,13] associative to Penning (σ_{as}/σ_{pe}) ratios^[14] and Penning Ionization Electron Spectra (PIES):^[10,13] crucially, all data have been measured under single collision condition and as a function of the collision energy. It has to be noted that the theoretical treatment, presented in this paper and stimulated by the PIES measurements in our laboratory, is able to rationalize in a unifying picture all the observables quoted above, probing different features of the optical potential (see eq. (3) in the main text). In particular, as demonstrated below, the adopted methodology is able to reproduce, in a state to state condition, all experimental data available for the Ne^*-Kr system obtained from our and other laboratories, including total and partial ionization cross sections and branching ratios between selected channels. Therefore, the proposed theoretical approach, which also includes within the same framework exchange and radiative mechanisms proposed in the past,^[9] is general and can be used to describe in a state to state condition the reactivity of all chemi-ionization reactions, including those involving molecules. A schematic view of the MB apparatus available in our laboratory is given in the left upper panel of Figure 1. PIES data and extracted peak area ratios reported in the same figure confirm that the reaction probability is dependent on the reaction channel, identified by the spin-orbit state of neutral reactants and ionic products, defined by the total electronic quantum number J , and it is modulated by the collision energy. Therefore, although an overview of the global observed phenomenology, including PIESs and many other observables, suggests the existence of important selectivity in the reaction dynamics, a complete and simultaneous rationalization of all experimental findings is still lacking.";

Furthermore, also the following sentences from page 11 (line 5 from the bottom) to page 12 (line 3 from the top) of the old manuscript version, have been moved to SI: "The complete formulation with the list of parameters is given in the main text: they have been obtained from a semi-empirical method founded on the ample phenomenology of the interactions of open-shell "P" atoms (particularly halogen

atoms). They have been investigated in detail with scattering experiments, performed with state selected atomic beams and analyzed with a proper theoretical treatment (see for instance ref. 15 and references therein). The Figure S2 depicts also some adiabatic electronic rearrangements and the configuration interaction between molecular states of the same symmetry coupled by CT components, which represent the main contributions to the interaction anisotropy in the adiabatic framework.”

In order to be more concise, in the “Introduction” section the following sentences have been deleted:

- at the end of page 3 of the old manuscript version: “they occur efficiently when the excited complex $(X\cdots M)^*$ exhibits a lifetime with respect to auto-ionization, shorter than the collision time, which is of the order of $\sim 10^{-12}$ s in the thermal energy range.”;
- at page 9 (line 3 from the top until the bottom page) of the old manuscript version: “All these criteria are here adopted to formulate explicitly a new methodology, which is applied for the first time to a prototype atom-atom reaction. In particular, X^* and M reactants are, respectively, a metastable $\text{Ne}^*(^3P_1)$ atom and the $\text{Kr}(^1S_0)$. Ne^* is excited in $^3P_{2,0}$ atomic sublevels, for which the outer electron is in the 3s orbital and the ionic core exhibits a $2p^5$ configuration. Therefore, at large separation distances Ne^* presents the same chemistry of sodium, while, at short distances, that is within the $(X\cdots M)^*$ collision complex (see picture in Figure 1), its ionic core manifests a strongly electron affinity, being iso-electronic with the fluorine atom. For atom-atom reactions, as the present one, also the ionic products are open-shell anisotropic-halogen like species. As mentioned above, an important validity test of the new approach, founded on the chemistry of mentioned species, regards its ability to rationalize in a unifying and compact picture the various experimental findings obtained in our and in other laboratories, with different experimental devices. They include also the results of recent and advanced MB experiments with state selected Ne^* beams,^[14] that provided for the first time the collision energy dependence of the $\sigma_{\text{as}}/\sigma_{\text{pe}}$ ratio on individual atomic sublevels of $\text{Ne}^*(^3P_2)$ reagent. The proposed investigation method also casts light on the dependence of the various experimental findings on specific features of the interaction both in entrance and exit channels. Moreover, the results here obtained suggest how to extend the methodology to auto-ionization reactions involving molecules, which represents an important future target of our research for their great relevance in several fields of applied sciences. Finally, our state to state theoretical approach is able to emphasize the electronic rearrangements within the transition state of many other chemical reactions that are more difficult to be fully characterized.”;
- at page 10 (lines 2-15 from the top) of the old manuscript version: “ As for many other systems, also in such simplified-prototype case the crucial point to be properly addressed concerns the formulation of W , including real and imaginary parts of the interaction represented in an internally consistent way (see eq. (3)). Specifically, it is important to note again that, while strength and anisotropy of the real part of W must depend on the different effectiveness of anisotropic adiabatic effects, strength and anisotropy of the imaginary part must be directly controlled by the selective balance of *non adiabatic* effects. As

stressed above, the latter cannot be considered fully independent of the adiabatic ones, since both are affected by rearrangements and changes in angular momentum couplings of valence electrons probed by the system during each collision event. General considerations, addressed to the understanding origin and features of *adiabatic* and *non adiabatic* effects, have been anticipated in recent papers.^[11] In the following, the guidelines for their proper application will be here fully provided and obtained results will be considered basic for an understanding and a general discussion of the selectivity in the chemical reactivity, explicitly including the dependence on the quantum states permitted.”;

- at page 12 (lines 5-10 from the top) of the old manuscript version: “They arise from rearrangements of valence electrons, promoted by polarization of the outer electronic cloud, changes in angular momentum couplings, spin-orbit and Coriolis effects^[11] (see also below). Crucial contributions come also from configuration interaction between molecular states of Σ and Π symmetry differing for one electron exchange (details on the correlation diagram between atomic and molecular states are given in SI). Therefore,”.

Furthermore, in order to allow the reader a better understanding of the paper, a new paragraph has been added to the revised manuscript moving the main part of the old SI with related references there. At the same time, the two old Figures S1 and S2 are now the new Figures 1 and 2 of the revised manuscript as requested by the reviewer. To allow this previous modification compatible with the text of the manuscript, we extracted the right lower panel of the old Figure 1 as a new Figure 6 that is inserted with the relative caption at page 22 of the revised version of the manuscript where the reported data are discussed. Such data need to be discussed in the main text of the manuscript since they are of importance being a further test of the validity of the proposed method which is additional to those indicated in Figs. 4 and 5. Furthermore, in our opinion is better to maintain the description of Cx and Cy coefficients since they are very important to allow a full understanding of the used method. The new added paragraph (see page 6 of the revised version of the manuscript) is the following:

“Experimental observables and interaction potential formulation

The proposed treatment has been stimulated by experiments performed with the molecular beam (MB) technique who provided total and partial ionization cross sections (σ)^{8,30}, associative to Penning (σ_{as}/σ_{pe}) ratios³¹⁻³⁴ and Penning Ionization Electron Spectra (PIES)^{25,30} the latter representing a sort of transition state spectroscopy. A schematic view of the molecular beam apparatus available in our laboratory and extensively described in previous papers^{27,28,35}, is given in the left upper panel of Supplementary Figure 1 of Supplementary Information (SI).

This section focuses on the formulation of the manifold of *adiabatic* potential energy curves, representative of V_i in the various reaction channels opened by atom-atom chemiionization processes, that account for all basic features of quantum states accessible to each investigated system. The proper identification and modeling of the leading interaction components involved is crucial to define also nature, strength, range and selectivity of *non adiabatic effects* determining the state to state I component. This is the crucial point to highlight and define all basic features and details of the state to state stereodynamics treatment applied to the investigated reactions. The following sections, reporting on the prediction of the scattering

quantities with their comparison with the experimental findings, provide a quantitative important test of the methodology. As for the analysis of many other microscopic properties the fundamental starting point concerns the formulation of the involved interaction which drives the collision dynamics.

For the representation of the real component V_t of the optical potential, including its dependence on the separation distance R and relative orientation of interacting partners, it is proper to assume that each entrance channel³⁵ is determined by the weighted sum of two limiting representations, whose relative role varies with R (major details are given in SI). However, it is useful emphasize that:

- a) For entrance channels, at large R , a simple neutral-neutral representation of the associated potential energy is sufficient since the system exhibits a substantial isotropic behavior typical of an alkaline atom interacting with a noble gas partner. An Improved Lennard Jones formulation of the interaction³⁶ represents completely this component³⁶ (see also SI for further details);
- b) At intermediate and short R , the anisotropic role of the ionic core of the metastable atom is emerging. Under these conditions the representation of the interaction in the entrance channels must take into account for the anisotropic contributions due open shell “P” nature of the ionic core^{37,38}.
- c) Also, in the exit channels the open shell nature of the atomic ion in “P” state controls the basic features of the involved interaction anisotropy.

Therefore, following the guidelines extensively developed in our laboratory^{38,39} it is possible to describe the interaction energy, when an open “P” shell atom or ion approaches a closed shell species, as an atom 1S_0 , by a manifold of potential energy curves, each one associated to a specific quantum state accessible to the system. The adopted representation, defined in terms of proper quantum numbers, accounts for all relative alignment (or orientation) of reagents and products permitted within the interatomic electric field which here becomes the proper quantization axis of the interacting system. Obtained curves must be considered effective *adiabatic* interaction components since including the contributions associated to pure Σ and Π molecular states, defined by the electronic quantum number $\Lambda=0$ and $\Lambda=1$ and identified as V_Σ and V_Π , mixed by spin orbit (SO) effects. It has been also demonstrated^{38,39} that for “P” atomic species it is sufficient to employ, for a full description of its anisotropic interaction with a 1S_0 atom partner, a weighted sum of V_0 and V_2 Legendre-expansion radial coefficients combined with the SO splitting. Specifically, the Legendre coefficients are defined as $V_0 = \frac{1}{3}(V_\Sigma + 2V_\Pi)$ and $V_2 = \frac{5}{3}(V_\Sigma - V_\Pi)$, while the inverse formulas are simply given by $V_\Sigma = V_0 + \frac{2}{5}V_2$ and $V_\Pi = V_0 - \frac{1}{5}V_2$. In this way, while V_0 describes the spherical average interaction component, all anisotropic contributions, are directly taken into account through the use of the anisotropic term V_2 . It is important to stress that the last term, accounting for the quantized spatial orientation of the valence orbitals of the open shell atom or ion within the interacting complex, is basic to describe the manifold of *adiabatic* potential energy curves, associated to all quantum states accessible, with their stability anisotropy. The case of 3P_J (or 2P_J) open shell atomic species, like Ne^* (or Ne^+) and Kr^+ , is characterized by a reversed sequence of spin orbit sublevels (see Fig. 1). Accordingly, effective *adiabatic* potential energy curves $V_{|J,\Omega\rangle}$ (J is the total electronic angular momentum quantum number, defining also the spin orbit level, while Ω

quantizes the absolute projection of the \mathbf{J} along R) have been defined for all channels and formulated as:

Entrance channels

$$\begin{aligned}
 V_{|0,0\rangle} &= V_0 + \frac{1}{10} V_2 + \frac{1}{2} \Delta_0 + \frac{1}{2} \left(\frac{9}{25} V_2^2 + \Delta_0^2 - \frac{2}{5} V_2 \Delta_0 \right)^{1/2} \\
 V_{|2,0\rangle} &= V_0 + \frac{1}{10} V_2 + \frac{1}{2} \Delta_0 - \frac{1}{2} \left(\frac{9}{25} V_2^2 + \Delta_0^2 - \frac{2}{5} V_2 \Delta_0 \right)^{1/2} \\
 V_{|2,1\rangle} &= V_0 + \frac{1}{10} V_2 + \frac{1}{2} \Delta_1 - \frac{1}{2} \left(\frac{9}{25} V_2^2 + \Delta_1^2 \right)^{1/2} \\
 V_{|2,2\rangle} &= V_0 - \frac{1}{5} V_2
 \end{aligned}$$

Exit channels

$$\begin{aligned}
 V_{|1/2,1/2\rangle} &= V_0 + \frac{1}{10} V_2 + \frac{1}{2} \Delta + \frac{1}{2} \left(\frac{9}{25} V_2^2 + \Delta^2 - \frac{2}{5} V_2 \Delta \right)^{1/2} \\
 V_{|3/2,1/2\rangle} &= V_0 + \frac{1}{10} V_2 + \frac{1}{2} \Delta - \frac{1}{2} \left(\frac{9}{25} V_2^2 + \Delta^2 - \frac{2}{5} V_2 \Delta \right)^{1/2} \\
 V_{|3/2,3/2\rangle} &= V_0 - \frac{1}{5} V_2
 \end{aligned}$$

where Δ_0 , Δ_1 and Δ are the energy splitting SO between fine atomic sublevels, identified by the quantum number J , whose definition and values are given in Fig. 1. As indicated above, in the entrance channels the V_0 term, coinciding with the isotropic component of $V_t(R)$, exhibits a mixed nature, accounting for the gradual passage, as R decreases, from neutral-neutral to ion-neutral system surrounded by an electron in a Rydberg (see left upper panel of Supplementary Figure 1). In the exit channels, a pure ion-neutral system operates. In the two cases, the V_0 terms have been represented by the ILJ model (see SI), suitable to describe pure non covalent interactions.

Fig. 1

The sequence of fine levels in entrance and exit channels for $\text{Ne}^*\text{-Kr}$ system. Note that, although $\text{Ne}^*(^3\text{P}_1)$ is not metastable and then it does not participate to the chemi-ionization reaction, the Δ_1 splitting is involved in the representation of the effective adiabatic potentials given in the text.

Moreover, for both entrance and exit channels, V_2 , that we have identified with the anisotropic configuration interaction between entrance and exit channels differing for one electron exchange (see Supplementary Figure 2 in SI), has been represented by an exponential decreasing function, defined by a pre-exponential factor A and an exponent α . This function, defined according to the guidelines reported elsewhere^{38,40} reflects the “canonical” dependence of the integral overlap between the atomic orbitals exchanging the electron. A further contribution $\frac{C_a}{R^6}$ due to the anisotropy of dispersion forces, has been also added. An important novelty of this method is that for entrance and exit channels the modulus of the exponential function is the same, while its sign is negative for exit and positive for entrance, since related to *bonding* and *antibonding* stabilization effects by charge transfer that arise from the configuration interaction between entrance and exit channels of the same symmetry^{37,38,40}, as depicted in Supplementary Figure 2 of SI. The additional contribution accounts for the role of polarizability anisotropy of the open shell species on asymptotic behavior of V_2 . All the potential parameters are given in Supplementary Table 1 of SI. The adoption of these criteria carries here to the potential energy curves consistent with the results from suitable theoretical methods on anisotropic interactions affected by the perturbation of weak chemical contributions^{41,42}.

Since the sign the exponential contribution to V_2 is positive for entrance and negative for exit channels, this interaction potential formulation leads to a different correlation between atomic states, representative of the system at long range separation distances, where $|V_2| \ll \Delta_i$, and molecular states of the same system emerging at short range, where $|V_2| \gg \Delta_i$.⁴¹

For major details on such correlation see Fig. 2 and Supplementary Figure 2 of SI.

CORRELATION DIAGRAM BETWEEN ATOMIC AN MOLECULAR STATES

ENTRANCE CHANNELS					EXIT CHANNELS			
Large R	Intermediate R	Short R	Σ character	Π character	Large-intermediate R	Short R	Σ character	Π character
atom-atom $ J,\Omega\rangle$ states	ion-atom $ J,\Omega\rangle$ states	molecular ion $2s+1\Lambda_n$ states			ion-atom $ J,\Omega\rangle$ states	molecular ion $2s+1\Lambda_n$ states		
$ 0,0\rangle$	$ 1/2,1/2\rangle$	$^2\Sigma_{1/2}$	C_x	$1-C_x$	$ 1/2,1/2\rangle$	$^2\Pi_{1/2}$	$1-C_y$	C_y
$ 2,0\rangle$	$ 3/2,1/2\rangle$	$^2\Pi_{1/2}$	$2/5 (1-C_x)$	$2/5 C_x$	$ 3/2,3/2\rangle$	$^2\Pi_{3/2}$	-	1
$ 2,1\rangle$	$ 3/2,3/2\rangle$	$^2\Pi_{3/2}$	$3/5 (1-C_x)$	$3/5 C_x + 1/5$				
$ 2,2\rangle$	$ 3/2,1/2\rangle$	$^2\Sigma_{1/2}$	-	$4/5$	$ 3/2,1/2\rangle$	$^2\Sigma_{1/2}$	C_y	$1-C_y$

Fig. 2

The correlation diagram between atomic and molecular states for $\text{Ne}^*\text{-Kr}$ system. In entrance and exit channels the sequence of molecular states is opposite because the bonding and antibonding effects due to the configuration interaction. *Entrance channel features:* The structure of the ionic adduct, surrounded by the excited electron in a Rydberg state (see text), is of relevance to determine crucial features of the system at intermediate and short R . *Entrance channels boundaries:* A large R , $C_x = 0.333$ and the two fine states $J = 2, 0$ of Ne^* exhibit Σ and Π character in the 1:2 statistical ratio. Moreover, the global weight (sum of both character degree) of $|2,0\rangle$ state is one half of that of $|2,1\rangle$ and $|2,2\rangle$, because of the different degeneracy. With the R decreasing, the $|2,1\rangle$ state assumes a Π molecular character faster than $|2,0\rangle$ since the spin orbit mixing is characterized by $\Delta_1 < \Delta_0$ (see Fig. 1). *Exit channels boundaries:* A large R , $C_y = 0.667$ for the $|3/2,1/2\rangle$ state and also in this case fine both levels exhibit Σ and Π character mixed in the 1:2 statistical ratio.

Moreover, it has been also deduced that the Σ and Π character of involved potential energy curves $V_{|J,\Omega\rangle}$ at all R values can be evaluated from the following relations⁴³:

$$\begin{aligned} V_{|2,0\rangle}, V_{|2,1\rangle}, V_{|3/2,1/2\rangle} &= \cos^2\alpha V_\Sigma + \sin^2\alpha V_\Pi \\ V_{|0,0\rangle}, V_{|1/2,1/2\rangle} &= \sin^2\alpha V_\Sigma + \cos^2\alpha V_\Pi \end{aligned}$$

where

$$\cos^2\alpha = \frac{1}{2} + \frac{\left(1 - \frac{9V_2}{5\Delta}\right)}{4\sqrt{2} \sqrt{1 + \left[\left(\frac{1 - \frac{9V_2}{5\Delta}}{2\sqrt{2}}\right)\right]^2}}$$

Again, in all cases only the exponential contribution to the V_2 component with its proper sign must be taken into the previous equation. This boundary arises from the meaning of the exponential term which is the only one selectively representing the configuration interaction anisotropy by charge transfer, which is the exclusive component determining the formation of molecular states, perturbed by *bonding* and *antibonding* effects. These formulas agree with the following asymptotic conditions

(see Fig. 2): at short distances, all potential energy curves must represent states having a pure Σ or Π molecular character, while at large distances, where SO coupling is dominant, a mixing of the molecular characters occurs. Note also that the adopted formulation of the interaction accounts for a different behavior of Ne^*-Kr (or Ne^+-Kr), with respect to $\text{Ne}-\text{Kr}^+$. The variation arises from the opposite sign of V_2 component and from the change in the role of SO mixing, defined by the different values of Δ_0, Δ_1 and Δ . The behavior of $V_{|2,2\rangle}$ and $V_{|\frac{3}{2},\frac{3}{2}\rangle}$ curves, effective in the entrance and exit channels, respectively, is not discussed in detail because they show at all distances a pure Π character. In the following, the adoption of C_x and C_y coefficients quantifies the Σ character degree in entrance and exit channels, respectively. Moreover, additional details on the adiabatic correlation between atomic and molecular states, both in entrance and exit channels, defined in terms of proper quantum numbers, are given in Fig. 2, where also the meaning of C_x and C_y coefficients, with their relevance in the definition of the suitable correlation between atomic and molecular states, is further justified. The relative role of Σ and Π molecular character in all entrance and exit channels can be also assessed from such a Figure.”.

The added references are the following:

- [35] B. G. Brunetti, P. Candori, S. Falcinelli, F. Pirani, F. Vecchiocattivi, *J. Chem. Phys.* **2013**, *139*, 164305.
- [38] F. Pirani, S. Brizi, L. F. Roncaratti, P. Casavecchia, D. Cappelletti, F. Vecchiocattivi, *Phys. Chem. Chem. Phys.* **2008**, *10*, 5489-5503.
- [39] V. Aquilanti, G. Liuti, F. Pirani, F. Vecchiocattivi, *J. Chem. Soc., Faraday Trans. 2*, **1989**, *85*, 955-964.
- [40] F. Pirani, A. Giulivi, D. Cappelletti, V. Aquilanti, *Mol. Phys.* **2000**, *98*, 1749-1762.
- [41] P. M. Dehmer, *Comments At. Mol. Phys.* **1983**, *13*, 205-227.
- [42] A. E. Reed, L. A. Curtiss, F. Weinhold, *Chem. Rev.* **1988**, *88*, 899-926.
- [43] P. Tosi, O. Dmitrijev, Y. Soldo, D. Bassi, D. Cappelletti, F. Pirani, V. Aquilanti, *J. Chem. Phys.* **1993**, *99*, 985-1003.

Reviewer #1 - addressed point 2: “One key issue is the semi-classical model used to simulate the experimental results portrayed in Figs. 4 and 5 which is the actual novelty of the paper. However, the authors state: ‘Although the fit of experimental data is outside the purpose of this research, which is exclusively addressed to an internally consistent prediction and rationalization of many different experimental findings ...’. Therefore, the reader is left without this information and without any reference where to find it.”

Author reply and made modifications: We agree with the reviewer #1. Indeed, the sentence indicated by the reviewer is not clear and generate confusion in the readers. For such a reason we modified the sentence just after the Fig. 4 (at page 20 line 6 from the bottom) in the revised version of the manuscript as it follows: “It is important to note that present semi-classical calculations are able to reproduce experimental results obtained in our and other laboratories since 1981^{33,45,46}. In Fig. 4, for comparison, some results of total cross section provided by Gregor and Siska⁴⁵, which are data averaged over all involved states of reagents and products, are reported.”

Reviewer #1 – addressed point 3: “As for the the reference list, I am afraid that it is not following the commonly accepted format: one reference for each paper. This has

some detrimental effects such as not knowing which of the articles included under a given reference (sometimes as many as ten) have to be preferably consulted by the interested reader. As an example, on Pg. 16, the text reads ‘...has been exploited to calculate for the first time, within a semi-classical treatment4...’, but, if the reader attempts to find which is the article suggested by the authors, he/she would find (to this/her dismay!) that there as many as 8 articles to peruse. In addition, there are many references that can be spared either for being too general or largely irrelevant for the content of the manuscript.”

Author reply and made modifications: We agree with the reviewer #1 and in order to address his comment we renumbered all references assigning a single number to each cited article. In such a reorganization we tried to avoid too generic references, limiting us in citing only articles strictly needed and inserting them in a more precise way to allow the reader a better and proper consultation. For such a reason the following references have been deleted:

- [1a] R. N. Zare, *Ber. Bunsenges Phys. Chem.* **1982**, 86, 422-425;
- [1c] S. Stolte, *Ber. Bunsenges. Phys. Chem.* **1982**, 86, 413-421;
- [1d] B. Friedrich, D. R. Herschbach, *Z. Phys. D-Atoms, Molecules and Clusters* **1991**, 18, 153-161;
- [1e] V. Aquilanti, M. Bartolomei, F. Pirani, D. Cappelletti, F. Vecchiocattivi, Y. Shimizu, T. Kasai, *Phys. Chem. Chem. Phys.* **2005**, 7, 291-300;
- [4a] F. M. Penning, *Naturwissenschaften* **1927**, 15, 818;
- [4b] F. M. Penning, *Z. Phys.* **1928**, 46, 335-348;
- [4c] V. Cermak, *J. Chem. Phys.* **1966**, 44, 3774-3780; 4
- [4d] A. Niehaus, *Ber. Bunsenges Phys. Chem.* **1973**, 77, 632-640;
- [5d] K. Ohno, *Bull. Chem. Soc. Japan* **2004**, 77, 887-908;
- [5e] M. Alagia, N. Balucani, P. Candori, *Rend. Fis. Acc. Lincei* **2013**, 24, 53-65;
- [6a] C. A. Arango, M. Shapiro, P. Brumer, *Phys. Rev. Lett.* **2006**, 97, 193202;
- [6c] M. Pawlak, Y. Shagam, A. Klein, E. Narevicius, N. Moiseyev, *J. Phys. Chem. A* **2017**, 121, 2194-2198;
- [7b] H. Conrad, G. Ertl, J. Küppers, W. Sesselmann, H. Haberland, *Surface Science* **1980**, 100, L461-L466;
- [7c] F. Bozso, J. T. Jr. Yates, J. Arias, H. Metiu, R. M. Martin, *J. Chem. Phys.* **1983**, 78, 4256-4269;
- [7e] S. Schokl, H. A. J. Meijer, M.-W. Ruf, H. Hotop, *Meas. Sci. Technol.* **1992**, 3, 544-551;
- [7f] D. Faubert, G. J. C. Paul, J. Giroux, M. J. Bertrand, *Int. J. Mass Spectrom. Ion Processes* **1993**, 124, 69-77;
- [7h] M. G. Ikonomou, S. Rayne, *Anal. Chem.* **2002**, 74, 5263-5272;
- [11c] S. Falcinelli, F. Pirani, P. Candori, B. G. Brunetti, J. M. Farrar, F. Vecchiocattivi, *Front. Chem.* **2019**, 7, 445.

Reviewer #1 – addressed point 4: “In summary, I cannot recommend the publication of this manuscript as it has been presented now. It is not self-readable or self-contained to be a review meant for illustrating the interested reader that may wish to approach the field. Many of the essential concepts to understand the manuscript are not explained at all or just in a cursory manner. The manuscript is neither a research article presenting new results (in spite of the claims by the authors); apart from Pgs. 16-20, the rest of the manuscript is written as a lengthy Introduction or anticipated Conclusions. I must say that I learnt more from ref. 11(a)

than from the content of this manuscript. As for the stereodynamical content, it is somewhat hidden in the text, and the authors do not make any attempt why that keyword is in the title. I should stress that I am not denying the scientific content of the manuscript, even less the sound scientific reputation of their authors. I am only stating, that in my opinion, for the manuscript to be recommended for publication must be rewritten thoroughly, possibly made more concise and, for sure, more self-contained.”

Author reply and made modifications: We understand the criticism expressed by the reviewer and we think that we fully addressed it by a deeply revision and rewrite of the manuscript as attested by the made modifications listed above. Furthermore, in order to explain to the reader the stereodynamical content of the manuscript that, as properly noted by the reviewer, is somewhat hidden in the text, we added the following sentences:

- at page 6, line 5 from the bottom of the revised version of the manuscript: “This section focuses on the formulation of the manifold of *adiabatic* potential energy curves, representative of V_i in the various reaction channels opened by atom-atom chemiionization processes, that account for all basic features of quantum states accessible to each investigated system. The proper identification and modeling of the leading interaction components involved is crucial to define also nature, strength, range and selectivity of *non adiabatic effects* determining the state to state Γ component. This is the crucial point to highlight and define all basic features and details of the state to state stereodynamics treatment for the investigated reactions. The following sections, reporting on the prediction of the scattering quantities with their comparison with the experimental findings, provide a quantitative important test of the methodology. As for the analysis of many other microscopic properties the fundamental starting point concerns the formulation of the involved interaction which drives the collision dynamics.”;
- At the end of page 14, line 11 from the bottom of the revised version of the manuscript: “The proposed interaction formulation is here fully exploited to evaluate state to state reaction probabilities for a prototype atom-atom process with their effects on the stereodynamics.”;
- in the revised text at page 26, line 2 from the top: “In particular, the ability of our method to describe the stereodynamics of chemi-ionization involving molecules in a state-to-state condition, should allow a correct modelling of interesting experimental data obtained by our^[4] and other laboratories^{12,15,47}.”, with the addition of the following reference:
[47] D. Watanabe, H. Ohoyama, T. Matsumura, T. Kasai, *J. Phys. Chem. A* **2007**, *111*, 30, 6915-6919.

Responses to the Reviewer #2 comments:

Reviewer #2 – addressed point 1: “This paper describes a general treatment of prototypical energy transfer reactions like Penning ionisation and related reactions. The method permits the extraction of orientation specific reactions and is applied to $Ne(^3P_{0,2})+Kr$ as an example. The theory is very promising and may help to greatly improve our understanding of elementary chemical reactions that play a role in many important environments. I recommend publication, but the manuscript deserves some attention before doing so. Overall, I suggest that the authors have a native-english

person proof-read the manuscript in order to eliminate all the smaller linguistic mistakes and make the paper overall more easily readable.”

Author reply and made modifications: We thank the reviewer #2 for his positive and encouraging comments and criticism. In order to improve the readability and the English style of the manuscript, we followed the suggestion of the reviewer via a proof-reading of the text by a native English colleague who allowed us to eliminate mistakes and improve the linguistic style (see the modifications made in the revised text and highlighted in red color).

Reviewer #2 – addressed point 2: *“Furthermore, I have the following specific comments: * Line 38: "Inter-phase" should probably be interface (even if the term inter-phase is actually quite nice).”*

Author reply and made modifications: In order to be more clear, at line 38 we replaced the sentence “at inter-phases” by “**between different phases**”;

Reviewer #2 – addressed point 3: *“* Line 46: I don't agree with the limitation of chemi-ionisation to reactions between excited species and something else. Chemi-ionisation reactions are also known between two ground state species.”*

Author reply and made modifications: In order to be precise in the definition of chemi-ionization reactions, at line 47 we added the following sentence: “**Such reactions, very common in nature, involve neutral species with sufficient energy to provide ionic products and are considered as the primary initial step in flames. They are mainly ...**”;

Reviewer #2 – addressed point 4: *“* Line 55: The left-going arrow with the electron in equation 2 is not clear.”*

Author reply and made modifications: In order to clarify the use of the left-going arrow in equation (2), at line 52 above eqs. (1) and (2), we added the following sentence: “**This is schematized by the left-going arrow in equation (2) below, representing the exchange of one electron from the collisional target M to the core of the metastable excited species X*:**”;

Reviewer #2 – addressed point 5: *“* In figure 2 (and others): The depiction of the p-orbitals in this fashion is a bit misleading and may make the reader think that the species are, in fact, oriented. While in some figures the orientation is essential, in others it is not, and the figure should probably be adapted.”*

Author reply and made modifications: We thanks the reviewer#2 for its proper suggestion. While in Fig, 3 the orbital orientation is essential, in Figs. 1 and 2 - that are moved in the SI as requested by the reviewer#1 (see above) becoming the new Figs. S1 and S2 - the depicted p-orbitals do not indicate a their mutual orientation but only the most probable orientation producing ionization. To specify this point, we added the following sentence in the caption of both Figs. S1 and S2: “**In the Figure the depicted p-orbitals do not indicate a specific and mutual orientation but only the most probable orientation producing ionization.**”;

Reviewer #2 – addressed point 6: *“* Line 266: (X...M) can be replaced by (Ne...Kr) since in the present context the text specifically discusses that system.”*

Author reply and made modifications: We agree with the reviewer#2 and at line 266 we replaced “(X...M)” with “(Ne...Kr)”;

Reviewer #2 – addressed point 7: “* page 15/16: It is not clear what is the relation of the Gamma terms with the ratio between the AI and PI cross sections. I believe that somewhere in the paper it would be worth mentioning how one makes the step from the potentials to cross sections, especially since the readership of this article may not be experts in scattering dynamics.”

Author reply and made modifications: We thanks the reviewer#2 for his proper suggestion. To address this point, we added the following sentence at page 23, line 5 from the top of the revised text: “In general, *direct* and *indirect* mechanisms tend to be favored, respectively, at small and high separation distance (mainly probed at high and low collision energy). From this crucial aspect it immediately emerges that strength and radial dependence of each state to state Γ component depend on a different critical balance of two mechanisms. It determines respectively, value and variation with the collision energy of the total ionization cross section, which represents the reaction probability of the selected channel. Moreover, it also suggests that the σ_{as}/σ_{pe} ratio mostly depends on the relative position of repulsive walls and wells in the adiabatic potential energy curves that control the dynamics, including trapping, in the entrance and exit channel of the selected state to state reaction (see Supplementary Figure 2 – left panel of SI).”.

Reviewer #2 – addressed point 8: “* Have the authors applied their model to the work by Kasai, Ohoyama, Watanabe et al.? This work should probably be mentioned in the paper at some point, since it experimentally investigates highly relevant systems as well.”

Author reply and made modifications: We haven't applied yet the proposed model to the work of Kasai and coworkers (*J. Phys. Chem. A* 2007, 111, 30, 6915-6919) but it could be a very interesting and further development in next future, since it reports on quite relevant data. This important future development is announced in the manuscript by inserting the following sentence in the revised text at page 25, line 2 from the top: “In particular, the ability of our method to describe the stereodynamics of chemi-ionization involving molecules in a state-to-state condition, should allow a correct modeling of interesting experimental data obtained by our^[4] and other laboratories^{12,15,47}.”, with the addition of the following new suggested reference: [47] D. Watanabe, H. Ohoyama, T. Matsumura, T. Kasai, *J. Phys. Chem. A* 2007, 111, 30, 6915-6919.

Responses to the Reviewer #3 comments:

Reviewer #3 – addressed point 1: “In this manuscript, the authors develop a new theoretical treatment - considering that entrance and exit channels form an unique manifold of coupled quantum states - to study state-to-state chemi-ionization reactions, and they apply it to the Ne^*-Kr system. Some parts of the manuscript are, in my opinion, quite confusing and very hard to follow, which makes me believe that the manuscript is too-specialized to warrant its publication in *Communications-Chemistry*. Perhaps, the manuscript could be publishable if the authors rewrite some parts of the manuscript, focusing on the novelty of their approach (what is new in their approach). In particular, I found very hard to understand how the computational data were calculated (in line 227 the authors say that "from a semi-empirical method" but the details are not found in the manuscript, and the authors only refer to Ref. 4 [8 articles]).”

Author reply: We thank the reviewer#3 for his proper suggestion and criticism which is analogous to the one expressed by the reviewer#1. To address this critical point, we completely revised our manuscript by a deep rewrite taking into account for a more general description of the used computational procedure to reproduce experimental data from our and other laboratories. In particular, we added a new paragraph titled “**Experimental observables and interaction potential formulation**” where the new used semi-empirical method is fully described allowing also a non-specialized reader to understand novelty and basic principles of the computational analysis applied. All modifications made in the revised version of the manuscript are fully reported in the “**Reviewer #1 – addressed points 1, 2 and 3**” paragraphs above, and in our feeling they should also fully satisfy the current point addressed by the reviewer#3.

Reviewer #3 – addressed point 2: *“It is also not clear for me what is the meaning of “anisotropy” in the contest of this manuscript. In line 85, the authors state that both components of W vary with R and with the relative orientation of the two atoms. However, a) it is not clear how the relative orientation is defined and, b) according to the Supporting Information, the real component only depends on R . On the same topic, it is also not clear the reduced variables on which the Legendre Polynomial depends.”*

Author reply: We thank the reviewer#3 for his suggestion. In order to be more clear, mentioning the anisotropy in the treated atom-atom autoionizing systems, we added the following sentences:

- at page 5, line 5 from the top of the revised manuscript: “**If in equations (1) and (2) M is an atom, the stereo-selectivity arises exclusively from the alignment kind of half filled orbitals within the interatomic electric field defined by quantized projections of electronic angular momentum.**”;
- at page 8, line 2 from the top of the revised manuscript: “**The adopted representation, defined in terms of proper quantum numbers, accounts for all relative alignment (or orientation) of reagents and products permitted within the interatomic electric field which here becomes the proper quantization axis of the interacting system.**”;
- at page 8, line 8 from the top of the revised manuscript: “**It has been also demonstrated^{38,39} that for “P” atomic species it is sufficient to employ, for a full description of its anisotropic interaction with a 1S_0 atom partner, a weighted sum of V_0 and V_2 Legendre-expansion radial coefficients combined with the SO splitting.**”

Reviewer #3 – addressed point 3: *“I found also quite confusing the discussion about the C_x and C_y coefficients. In line 244 the authors state that the Pi character degree is obtained as complement to 1 of the Sigma one (given by C_x and C_y for the entrance and exit channels, respectively). However, in Fig. 3 $1-C_x$ and $1-C_y$ are also related to Sigma character...”*

Author reply: We thank the reviewer#3 for his proper suggestion and criticism and to address this critical point in the attempt to clarify our discussion about C_x and C_y coefficients and Σ and Π characters, we added the following sentences:

- i) at page 13, line 13 from the top of the revised version of the manuscript and referred to Fig.2: “The relative role of Σ and Π molecular character in all entrance and exit channels can be also assessed from such a Figure.”;
- ii) at page 14, line 8 from the top of the revised version of the manuscript: “This treatment includes in the proper definition of V_0 term of entrance channel the role of polarization effects, while C_x and C_y coefficients are the proper marks of the transition from atomic to molecular states. In particular, the R regions where C_x and C_y are fast varying, the decoupling of electronic angular momenta is effective and *non adiabatic* effects become more probable.”;
- iii) at page 15, top line of the revised version of the manuscript: “According to the definition given above, C_x represents the Σ molecular symmetry degree, associated to a specific atomic orbital alignment, in the entrance channels and C_y the corresponding Σ molecular symmetry degree in the exit channels.”;
- iv) at page 15, line 5 from the top of the revised version of the manuscript: “... each one identified by J and Ω referring to Ne^+-Kr and $\text{Ne}-\text{Kr}^+$, being the states of the system effectively coupled by the charge transfer. Note that the different behavior arises from the opposite effect of the configuration interaction. Moreover, taking into account that in all cases the Π character degree is obtained as complement to 1 of the Σ one and that the states $|3/2,3/2\rangle$ exhibits a pure Π character, it appears that the $\Sigma:\Pi$ character ratio is 1:2 at all R according to their different degeneration. In addition, Fig. 2 provides all guidelines to obtain the Σ to Π character ratio for each entrance channel.”.

REVIEWERS' COMMENTS:

Reviewer #1 (Remarks to the Author):

Please, find attached report.

Reviewer #2 (Remarks to the Author):

I thank authors for taking into consideration my suggestions. I think that the manuscript has benefitted from all changes and is now almost ready for publication.

The only detail where I still do not agree with authors is about definition of chemi-ionisation as process that happens between excited species and other species that are ionised. Chemi-ionisation also happens between ground-state reactants, as long as the product cation has lower energy than the reactants. Many such processes are known, and reactions described here are just a special case - very common special case, but still not general case of chemi-ionisation.

Reviewer #3 (Remarks to the Author):

The authors have addressed my concerns, and accordingly, I recommend the manuscript for its publication.

Ref. Ms. No.: COMMSCHEM-20-0037A

Title: *A new general treatment describing the stereo-dynamics of state-to-state chemi-ionization reactions as prototype of gas phase oxidation processes.*

Authors: Stefano Falcinelli, Franco Vecchiocattivi¹ and Fernando Pirani.

The authors have undertaken substantial changes in the manuscript addressing the recommendations and criticisms of the reviewers. I am fairly content with the science of paper but I have some criticisms and suggestions about how the paper is laid out. I am aware that the format of Comm. Chem. imposes some limitations (only two sections: Introduction and Results and Discussion) but I would suggest the authors some changes to ease the reading of the manuscript:

1. The new last paragraph in the Introduction that starts with “In conclusion...” would ne probably better place in the end of the Results and Discussion section.
2. A sort of brief summary of the subject that will be exposed in the Results and Discussion section may be timely.
3. Until Pg. 8 there is no indication that the paper is going to deal with $\text{Ne}^* + \text{Kr}$ collisions. Even if the treatment is aimed to be general, the results are exclusively devoted to that process.
4. Rather surprisingly, after dealing with $\text{Ne}^* + \text{Kr}$ for 6 pages, it appears a subsection: The $\text{Ne}^* - \text{Kr}$ case. This and the next subsection (State-to-state cross sections and branching ratios) are probably the most interesting part of the article although most of the results have been presented in other publications by the authors. The latter subsection, of course deals with the $\text{Ne}^* - \text{Kr}$ system.

In summary, although I have no complains about the scientific content, I am afraid that I still find the manuscript excessively long, not well structured, and as a result of that, difficult to be read especially when it is intended to be published in a journal with general chemistry scope. Of course, this is my opinion that may differs from that of the authors. Still, I would encourage the authors to read again the manuscript and perhaps rewrite some parts with two main purposes: to shorten it and to make it more friendly to the reader.

Apart from this, I have no other objections for its publication.

Point-by-point responses to the reviewers' comments - manuscript COMMSCHEM-20-0037A: second revision.

We thank all reviewers #1, #2 and #3 as well as the Editor for their further comments and suggestions aimed to improve the quality of the manuscript COMMSCHEM-20-0037A.

We made a second revision our manuscript, addressing all suggestions by the reviewers.

All made changes in the manuscript are reported in red color in its revised version (see new uploaded revised manuscript) and are listed below.

Responses to the Reviewer #1 comments:

Reviewer #1 – addressed point 1: *“I am aware that the format of Comm. Chem. imposes some limitations (only two sections: Introduction and Results and Discussion) but I would suggest the authors some changes to ease the reading of the manuscript:*

1. The new last paragraph in the Introduction that starts with “In conclusion...” would be probably better place in the end of the Results and Discussion section.”

Author reply and made modifications: We thank the reviewer #1 for his comment and we agree with him and we moved the indicated following paragraph at the end of the second section (see page 26, line 526): *“Moreover, the emphasized microscopic mechanisms, driven by valence electron polarization, changes in electron angular momentum couplings and by selective charge transfer effects, all operative within the collision complex, are of great relevance from a point of view of fundamental science. Specifically, they help to identify the role of electronic rearrangements determining the fate of the TS of gas-phase oxidation reactions as well as many other types of chemical processes that are more difficult to characterize. Indeed, important redox reactions occur in liquid phase with a complex multi-steps mechanisms where reagents, products and exchanged electrons are solvated species. Instead, chemi-ionizations are elementary one-step gas phase reactions, involving the exchange of one single electron and this allowed us to fully characterize the electron rearrangements inside the TS with a high-level detail.”;*

Reviewer #1 - addressed point 2: *“2. A sort of brief summary of the subject that will be exposed in the Results and Discussion section may be timely.”*

Author reply and made modifications: We agree with the reviewer #1. Hence, at the end of the “Introduction” section we added the new following sentence, according to the reviewer#1 suggestion: *“The next section focuses on the W formulation adopting suitable-operative relations for V_t and Γ components which properly include their interdependence. This allowed us to develop a theoretical treatment, applicable even under state-to state reaction conditions, characterizing the electron rearrangements inside the TS with a high-level detail. The proposed methodology is applied to the prototype Ne^* -Kr atom-atom reaction, for which several experimental findings are available (its extension to other cases is in progress). Important validity tests are performed by comparing experimental data with predicted values of observables.”;*

Reviewer #1 – addressed point 3: *“Until Pg. 8 there is no indication that the paper is going to deal with $Ne^* + Kr$ collisions. Even if the treatment is aimed to be general, the results are exclusively devoted to that process.”*

Author reply and made modifications: The reviewer #1 is right and have added the following sentence at the end of the “Introduction” section at page 6, line 2 from the top: *“The proposed methodology is applied to the prototype $Ne^* - Kr$ atom-atom reaction, for which several experimental findings are available (its extension to other cases is in progress)”*;

Reviewer #1 – addressed point 4: *“Rather surprisingly, after dealing with $Ne^* + Kr$ for 6 pages, it appears a subsection: The $Ne^* - Kr$ case. This and the next subsection (State-to-state cross sections and branching ratios) are probably the most interesting part of the article although most of the results have been presented in other publications by the authors. The latter subsection, of course deals with the $Ne^* - Kr$ system.”*

Author reply and made modifications: Regarding this point, we feel that the modification mentioned in the previous “addressed point 3” could be enough to satisfy this comment.

Reviewer #1 – final comment: *“... I am afraid that I still find the manuscript excessively long, not well structured, and as a result of that, difficult to be read especially when it is intended to be published in a journal with general chemistry scope. Of course, this is my opinion that may differ from that of the authors. Still, I would encourage the authors to read again the manuscript and perhaps rewrite some parts with two main purposes: to shorten it and to make it more friendly to the reader...”*

Author reply and made modifications: The reviewer#1 demonstrated a very careful and detailed peer-review of the manuscript. We are very grateful to him and we tried to address this last comment. In order to better organize the manuscript and to improve its readability, the changes we have made and which are listed above seem appropriate to us. Furthermore, we have carefully re-read the manuscript in order to be more clear and concise as well as to improve the English style. The main modifications that we made in our manuscript are reported in red color in the revised text, and are listed as it follows:

- At page 4, lines 76-77, the following sentence has been deleted: *“It is here applied to a prototype atom-atom reaction, and we are planning its extension to other cases.”*;
- At page 5, lines 105-106, the following sentence has been deleted: *“parts of eq. (3) (illustrated in the next section through the adoption of suitable-operative relations)”*;
- At page 6, lines 125-128, the old sentence has been reworded as it follows: *“The proposed treatment has been stimulated by experimental findings from collision energy dependence measurements of total and partial ionization cross sections (σ)^{8,30}, associative to Penning (σ_{as}/σ_{pe}) ratios³¹⁻³⁴ and Penning Ionization Electron Spectra (PIES)^{25,30,35} the latter representing a sort of TS spectroscopy.”*;
- At page 6, lines 128-131, the following sentence has been moved in the “Method” section: *“A schematic view of MB apparatus available in our*

laboratory and extensively described elsewhere^{27,28,35}, is given in the Supplementary Methods and Supplementary Fig. 1.”;

- At page 7, lines 135-139, the old sentence has been reworded as it follows: “Identification and modeling of the leading interaction components involved permit to define nature, strength, range and selectivity of *non adiabatic effects* determining the state to state Γ component. This is the crucial point to highlight and define all basic features of the state-to-state stereodynamics treatment.”;
- At page 7, lines 141-143, the following sentence has been deleted: “As for the analysis of many other microscopic properties the fundamental starting point concerns the formulation of the involved interaction which drives the collision dynamics.”;
- At page 8, lines 169-171, the old sentence has been reworded as it follows: “For “P” atomic species it is sufficient to employ, for a full description of its anisotropic interaction with a 1S_0 atom partner^{38,39},...”.

Responses to the Reviewer #2 comments:

Reviewer #2 – addressed point: “... *The only detail where I still do not agree with authors is about definition of chemi-ionisation as process that happens between excited species and other species that are ionised. Chemiionisation also happens between ground-state reactants, as long as the product cation has lower energy than the reactants. Many such processes are known, and reactions described here are just a special case - very common special case, but still not general case of chemi-ionisation.*”

Author reply and made modifications: We thank the reviewer #2 for his very scrupulous clarification. In our manuscript the definition of Chemi-ionization that we have used comes from the IUPAC definition that says: “[Chemi-ionization] refers to a process whereby gaseous molecules are ionized when they interact with other internally excited gaseous molecules or molecular moieties.”

(doi.org/10.1351/goldbook.C01044). We could even agree with the referee and, in order to consider its suggestion, we can insert the following sentence in the “Introduction” section, at page 3, lines 45-49) modifying the previous one: “Chemi-ionization reactions are very common in nature. They occur when neutral reagents promote the formation of most stable ionic products and are then considered as primary steps in flames. When these processes involve an internally excited gaseous reagent species, they are also named as collisional autoionizations, or Penning ionization reactions. Such reactions can be considered as prototype of gas phase bimolecular oxidation processes occurring at thermal energy, and are mainly triggered...”.